# The Heteromeric Dopamine Receptor D2:D3 Controls the Gut Recruitment and Suppressive Activity of Regulatory T-Cells

**DOI:** 10.3390/ijms262010069

**Published:** 2025-10-16

**Authors:** Jacob Mora, Iu Raïch, Valentina Ugalde, Gemma Navarro, Carolina Prado, Pia M. Vidal, Pedro Leal, Alexandra Espinoza, Moting Liu, Rinse Weersma, Ranko Gacesa, Marcela A. Hermoso, Rafael Franco, Rodrigo Pacheco

**Affiliations:** 1Centro Científico y Tecnológico de Excelencia Ciencia & Vida, Fundación Ciencia & Vida, Avenida Del Valle Norte #725, Huechuraba, Santiago 8580702, Chile; jmoram@correo.uss.cl (J.M.); vugalde@cienciavida.org (V.U.); cprado@cienciavida.org (C.P.); pedro.leal.ftv@gmail.com (P.L.); marialexaesp@gmail.com (A.E.); 2Facultad de Medicina, Universidad San Sebastián, Providencia, Santiago 7510156, Chile; 3Departmento de Bioquímica y Fisiología, Facultad de Farmacia y Ciencia de los Alimentos, Universidad de Barcelona, 08028 Barcelona, Spain; iuraichipanisello@gmail.com (I.R.); g.navarro@ub.edu (G.N.); 4Fundación Arturo López Pérez OECI Cancer Center, Providencia, Santiago 7500921, Chile; 5Centro de Investigación en Red, Enfermedades Neurodegenerativas (CiberNed), Instituto de Salud Carlos III, 28029 Madrid, Spain; rfranco@ub.edu; 6Neuroimmunology and Regeneration of the Central Nervous System Unit, Biomedical Science Research Laboratory, Basic Sciences Department, Faculty of Medicine, Universidad Católica de la Santísima Concepción, Concepción 4080000, Chile; pvidal@ucsc.cl; 7Department of Gastroenterology and Hepatology, University Medical Center Groningen, 9713 GZ Groningen, The Netherlands; m.liu@umcg.nl (M.L.); r.k.weersma@umcg.nl (R.W.); r.gacesa@umcg.nl (R.G.); m.a.hermoso@umcg.nl (M.A.H.); 8Laboratory of Innate Immunity, Program of Immunology, Institute of Biomedical Sciences, Faculty of Medicine, Universidad de Chile, Santiago 8331150, Chile; 9Departmento de Bioquímica y Biomedicina Molecular, Facultad de Biología, Universidad de Barcelona, 08028 Barcelona, Spain; 10Facultad de Ciencias, Universidad San Sebastián, Providencia, Santiago 7510156, Chile

**Keywords:** dopamine receptors, regulatory T-cells, lymphocyte migration

## Abstract

Since colonic dopamine levels are markedly reduced during inflammatory bowel disease (IBD), we investigated how dopamine affects regulatory T-cells (Treg), which critically limit gut inflammation. Previously, we showed that the stimulation of the high-affinity dopamine receptor D_3_ (Drd3) impairs suppressive Treg activity and limits their recruitment into the colon upon gut inflammation. Here we study the role of the low-affinity dopamine receptor Drd2 in Treg. We find that mice harbouring Drd2-deficient T-cells developed more severe colitis induced by dextran sodium sulphate. The stimulation of Drd2 potentiated the suppressive Treg activity and increased their ability to reach the colonic tissue. A transcriptomic analysis of intestinal mucosa from IBD patients revealed an association with increased *DRD3* and reduced *DRD2* expression. Bioluminescence resonance energy transfer assays revealed that Drd2 and Drd3 form a heteromer. An in situ proximity ligation assay indicated that the Drd2:Drd3 heteromer was expressed on colonic Treg, and its expression was increased upon inflammation. Using peptides analogous to the transmembrane (TM) segments from Drd2 and Drd3 in bimolecular fluorescence complementation assays, we found TM peptides able to disassemble this heteromer. The heteromer disassembly dampened the suppressive Treg activity and impaired the recruitment of Treg into the colon upon inflammation. Our findings indicate that the Drd2:Drd3 heteromer constitutes a dopamine sensor that regulates suppressive Treg activity and their colonic recruitment.

## 1. Introduction

Inflammatory bowel diseases (IBDs) form a group of chronic remittent inflammatory disorders of the gastrointestinal tract, among which Crohn’s disease (CD) and ulcerative colitis (UC) are the most common. Evidence from mouse models of inflammatory colitis and samples obtained from CD and UC patients have indicated that gut inflammation in IBD is driven mainly by the inflammatory effector CD4^+^ T-cell (Teff) subsets T-helper 1 (Th1) and Th17 [1,2]. In addition, regulatory T-cells (Treg), a suppressive subset of lymphocytes playing a crucial role in maintaining intestinal homeostasis in healthy conditions, seems to be dysfunctional in IBD [3]. The transfer of exogenous Treg cells suppresses inflammation induced by Th1 and Th17 lymphocytes in mouse models of inflammatory colitis [4,5,6], and the main suppressive mechanism relies on interleukin (IL)-10 secretion by these cells. Indeed, the specific ablation of the *il10* gene in Treg (Foxp3^+^ CD4^+^) cells results in spontaneous colitis, highlighting the fact that IL-10 produced by Treg is fundamental in maintaining tolerance, particularly in intestinal tissues [7]. Treg cells are increased in the inflamed lamina propria of IBD patients in comparison to non-inflamed mucosa and mucosa from healthy controls. After isolation, these cells retain their ability to suppress effector T-cells in vitro [8,9], thus suggesting that the suppressive activity of Treg may be attenuated only in situ by mediators produced by the inflamed gut mucosa.

The marked decrease in dopamine levels in the inflamed gut mucosa from CD and UC patients [10] may affect the function of immune cells expressing dopamine receptors, including Treg and Teff. Importantly, reduced levels of intestinal dopamine have also been observed in inflamed gut mucosa using animal models of inflammatory colitis [11,12]. Dopamine exerts its effects by stimulating dopamine receptors, termed Drd1-Drd5, all belonging to the superfamily of G-protein coupled receptors (GPCRs) [13]. It is important to consider that each dopamine receptor displays different affinities for dopamine, with Ki values (in nM) of 27, 228, 450, 1705, and 2340 for Drd3, Drd5, Drd4, Drd2, and Drd1, respectively; thereby their stimulation depends on dopamine levels [12]. Our previous studies showed that *Drd3*-deficient naive CD4^+^ T-cells display impaired Th1 differentiation and reduced Th17 expansion [14,15]. The reduction in intestinal dopamine levels (from ≈1000 nM in healthy individuals to ≈100 nM in CD and UC patients [10,16]), and the fact that Drd3 may be selectively stimulated at low dopamine concentrations [17], suggest that low dopamine levels in inflamed gut mucosa favour the inflammatory potential of CD4^+^ T-cells, thus promoting gut inflammation. Accordingly, our previous work has shown that *Drd3* deficiency in CD4^+^ T-cells results in attenuated inflammatory colitis in mice [14].

Emerging evidence from several animal models of inflammation indicates that high dopamine levels exert a strong anti-inflammatory effect by stimulating low-affinity dopamine receptors, including Drd1 and Drd2 [18,19,20]. In this regard, high dopamine concentrations in the gut of healthy individuals would stimulate Drd1 in macrophages, attenuating the activation of the inflammasome NLRP3 and thereby abrogating the production of inflammatory cytokines [19]. Moreover, high dopamine levels would promote DRD2 stimulation, thus aiding the production of the anti-inflammatory cytokine IL-10 by human CD4^+^ T-cells in vitro [21] and also reducing both increased motility and ulcer development in an animal model of intestinal lesions [22]. Indeed, the genetic polymorphism of the *DRD2* gene includes an allele that involves decreased receptor expression, a risk factor for IBD [23]. Accordingly, although the frequency of Treg cells did not change in the gut, the suppressor function of intestinal Treg cells was compromised in inflammatory colitis [3], a condition associated with decreased dopamine levels [11]. Moreover, the impairment of suppressive Treg function was abolished by the administration of cabergoline, a Drd2 agonist [3,24]. Thus, collectively these findings suggest that Drd2 signalling in Treg cells promotes suppressive function in a healthy gut mucosa containing high dopamine levels.

Gut-homing of T-cells is a fundamental process to maintain tolerance to food- and microbiota-derived antigens. CD103^+^ dendritic cells (DCs) capture antigens coming from the gut lumen, process them and migrate from the intestine into mesenteric lymph nodes (MLNs) and Peyer’s patches (PPs), where they present these antigens to naive CD4^+^ T-cells [25,26,27]. Differently from DCs from other sources, CD103^+^ DCs coming from the gut express retinaldehyde dehydrogenase 2, which allows them to synthesise retinoic acid (RA) using the vitamin A captured from the gut [26]. By producing RA, CD103^+^ DCs induce the up-regulation of the chemokine receptor 9 (CCR9) and the integrin α4β7 in activated CD4^+^ T-cells, thus imprinting the ability to be recruited into the intestinal tissue (gut tropism) to these cells. Thereby, in the absence of inflammatory cues, CD103+ DCs arriving to MLNs and PPs present antigens to antigen-specific naive CD4+ T-cells, inducing their differentiation into Treg cells with gut tropism [27]. Subsequently, these Treg cells are recruited into the gut by CCL25 (a chemokine produced by endothelial cells in gut mucosa, corresponding to the ligand for CCR9) and MadCAM-1 (Mucosal vascular addressing Cell Adhesion Molecule 1, a surface molecule expressed by mucosal venules, which corresponds to the ligand of α4β7 integrin) to infiltrate into the gut lamina propria. Once Treg cells infiltrate the gut mucosa, they are exposed to IL-10, a cytokine produced constitutively by mucosal homeostatic CD11c+ macrophages [28]. This step is required to confer IL-10-producing capacity to Treg, a critical function for the generation of oral tolerance [27]. Our previous study showed that the heteromer formed by CCR9 and Drd5 is the surface receptor that provides gut tropism to Teff (but not to Treg) upon gut inflammation [29,30]. Thus, this data indicates that dopaminergic signalling plays an important role in T-cell migration upon gut inflammation.

Our previous study shows that genetic *Drd3* deficiency results in attenuated disease manifestation in two mouse models of inflammatory colitis, which was associated with increased IL-10 production by Treg cells infiltrating the colonic lamina propria. Furthermore, *Drd3* deficiency enhanced Treg colonic tropism, favouring their infiltration into the colonic mucosa upon intestinal inflammation [31]. Accordingly, *Drd3* deficiency in Treg cells exacerbated their therapeutic potential in vivo when transferred into wild-type mice undergoing inflammatory colitis [31]. Here we aimed to study the role of Drd2 in Treg in gut inflammation. Interestingly, compared with Drd3-mediated effects, our results indicate that Drd2 signalling induces an opposite effect on the suppressive activity and gut tropism in Treg. Unexpectedly, we found a novel molecular regulator of Treg function in the intestine, a heteromeric complex formed by Drd2 and Drd3 that acts as a dopamine sensor triggering different biological effects on Treg depending on the levels of dopamine. Moreover, our results suggest that the reduction in dopamine levels associated with gut inflammation and the consequent shift in dopamine receptors stimulated in the Drd2:Drd3 heteromer expressed on mucosal Treg cells represents one of the molecular changes responsible for the Treg unresponsiveness observed in the gut mucosa of IBD patients.

## 2. Results

### 2.1. Drd2-Mediated Signalling Favours the Suppressive Activity of Treg and Limits Gut Inflammation

To address the role of Drd2 signalling in CD4^+^ T-cells in the context of gut inflammation, we used a mouse model of inflammatory colitis induced by the administration of dextran sodium sulphate (DSS) in the drinking water, as in our previous study [31]. Since we hypothesised that Drd2 signalling in CD4^+^ T-cells is anti-inflammatory, we analysed the disease manifestation in mice harbouring *Drd2*-deficient CD4^+^ T-cells (*Drd2^flox/flox^*/*CD4^Cre^* mice, here called *Drd2^f/f^*/*CD4^Cre^*) using a suboptimal concentration of DSS (1%). The disease severity was quantified as the extent of the loss of initial body weight. As expected, the results show that *Drd2^f/f^/CD4^Cre^* mice developed a more severe disease manifestation when compared with mice harbouring *Drd2*-sufficient CD4^+^ T-cells (*Drd2^f/f^*, control littermates) (Figure 1A). To test whether this exacerbated inflammatory response was related to a lower Treg response in *Drd2^f/f^/CD4^Cre^* mice, we next analysed the production of IL-10 by CD4^+^ T-cells associated with the intestine. To this end, cells isolated from the MLNs of these mice were restimulated ex vivo and the extent of IL-10 production by CD4^+^ T-cells was quantified by intracellular cytokine staining followed by flow cytometry analysis. The results show that the *Drd2* deficiency in CD4^+^ T-cells results in a significant reduction in the IL-10 production by these cells under inflammatory conditions (Figure 1B).

To gain more robust evidence on the anti-inflammatory role of Drd2-mediated signalling in vivo, we aimed to complement the genetic evidence with a pharmacologic approach. For this purpose, wild-type mice were treated with a single i.p. injection of a selective Drd2 agonist (sumanirole) or only a vehicle (as a control) and then exposed to optimal concentrations of DSS (1.75%). The results show a significant attenuation of the body weight loss in those mice receiving sumanirole (Figure 1C), although there were no differences in the colonic histological score (Figure 1D). Altogether, these results suggest therapeutic potential for the pharmacologic stimulation of Drd2 in the context of inflammatory colitis.

All these results suggest that Drd2 stimulation of CD4^+^ T-cells dampens their inflammatory function and favours the production of the anti-inflammatory cytokine IL-10 (Figure 1A–C). To explore whether Drd2-mediated signalling directly impacts Treg activity, in vitro suppressive assays were undertaken. Accordingly, Treg were stimulated with a Drd2 agonist or vehicle in vitro and then the ability to suppress the proliferation of naive CD4^+^ T-cells was quantified. The results revealed that Drd2 stimulation substantially improves Treg suppressive activity (Figure 1E). Altogether these results provide genetic and pharmacologic evidence indicating that Drd2-mediated signalling enhances the ability of Treg to reduce gut inflammation.

### 2.2. Drd2-Mediated Signalling Promotes Colonic Tropism of Treg upon Gut Inflammation

Since we observed opposite biological effects of Drd2 (Figure 1) and Drd3 [31] on the suppressive Treg activity, we speculated whether these divergent effects extended to the regulation of gut tropism [31]. Accordingly, we used an in vivo migration assay in which congenic Treg cells isolated from mice harbouring *Drd2*-deficient (KO; CD45.2^+^) or *Drd2*-sufficient (WT; CD45.1^+^) CD4^+^ T-cells were i.v. transferred into wild-type congenic recipient mice (CD45.1^+^ CD45.2^+^) previously exposed to optimal DSS concentrations for 72 h. After 48 additional hours the arrival of donor Treg was quantified in the colonic lamina propria and the spleen (as a control). The results revealed a substantial and selective reduction on the arrival of *Drd2*-deficient Treg into the colonic lamina propria without changes in the ability of these cells to reach the spleen (Figure 2A).

Since our previous work indicated that Drd3-mediated signalling reduced the surface CCR9 expression on Treg and consequently dampened their arrival to the inflamed colonic mucosa [31], which release CCL25 [32], we next determined the effect of Drd2 on CCR9 expression and on CCL25-induced migration. To this end, we obtained Treg cells from MLNs of mice harbouring *Drd2*-deficient or *Drd2*-sufficient CD4^+^ T-cells and compared their migratory ability towards CCL25 in transwell assays. Whereas the migration of *Drd2*-sufficient Treg was significantly increased by CCL25, this chemokine had no effect on the extent of migration of *Drd2*-deficient Treg (Figure 2B). In agreement with the loss of CCL25-induced migration on *Drd2*-deficient Treg, we detected a significant reduction on the surface CCR9 expression of these cells (Figure 2C). Overall, these results indicate that Drd2-mediated signalling on Treg is required to induce a proper CCR9 expression and the consequent migration to CCL25, facilitating their recruitment into the colonic lamina propria upon gut inflammation.

### 2.3. IBD Patients Display Increased DRD3 and Reduced DRD2 Expression in Mucosal Lesions

Since previous results show Drd2 signalling favouring the colonic tropism of Treg and their suppressive activity compared with Drd3 stimulation limiting gut tropism and the suppressive activity of Treg [31], we questioned whether the expression of these receptors was altered in IBD patients. Accordingly, we compared the DRD2 and DRD3 transcript levels obtained from the intestinal mucosa of IBD patients and healthy controls (HCs). The bulk RNA sequencing data of intestinal biopsies from HC, CD and UC patients [33] showed a reduced expression of *DRD2* in the inflamed colonic tissue of UC patients compared with the flanking non-inflamed tissue and HC tissue (Figure 3). No differences in *DRD2* expression were found in the ileum. Remarkably, the inflamed colonic mucosa and flanking areas from CD or UC patients expressed higher levels of *DRD3* compared with HCs (Figure 3). *DRD3* transcript levels were also increased in the inflamed ileum mucosa of IBD patients compared with HCs (Figure 3). Altogether, these results reveal that gut inflammation in IBD is associated with an increase in *DRD3* expression and a reduction in *DRD2* expression, especially in the colonic mucosa.

### 2.4. Drd2 and Drd3 Form a Heteromeric Complex on Colonic Treg Cells

Since Drd2 and Drd3 signalling exerts opposite biological effects at the level of suppressive activity and the colonic tropism of Treg, we suspected that such a crosstalk might be mediated by a receptor heteromer formation. To address this possibility, we transfected HEK293T-cells with a constant amount of cDNA coding for Drd3 fused to RLuc (Drd3-RLuc) and increasing amounts of cDNA coding for Drd2 fused to YFP (Drd2-YFP) and conducted bioluminescence resonance energy transfer (BRET) assays in the presence of a luciferase substrate. The results revealed a BRET saturation curve (Figure 4A), indicating a specific interaction between Drd2 and Drd3. To determine whether the Drd2:Drd3 interaction is selective, we performed similar experiments, by transfecting the adenosine receptor A_1_ (A_1_R) fused to RLuc (A_1_R-RLuc) instead Drd3-RLuc, and we observed no significant BRET, indicating no interaction between A_1_R and Drd2 (Figure 4A).

To analyse the crosstalk of Drd2 and Drd3 at the level of signalling pathways activated, we determined the ability of stimulating these receptors to modulate cAMP production. Both Drd2 and Drd3 stimulation were coupled with the inhibition of cAMP production, and the simultaneous stimulation synergies on it (Appendix A). In addition, we analysed the potential coupling of Drd2 and Drd3 to the activation of the MAPK and AKT pathways. The results show that both Drd2 and Drd3 induced the phosphorylation of ERK1/2 and of AKT; however their effects were not additive (Appendix A).

Afterwards, we aimed to determine the transmembrane (TM) segments from Drd2 and from Drd3 involved in this interaction. We designed α-helix peptides with an analogous sequence to the Drd2 or Drd3 TM domains, coupled to a cell-penetrating peptide derived from the transactivator of transcription (TAT) protein of the human immunodeficiency virus, giving proper orientation to each peptide within the plasma membrane (Table 1). These TAT-TM peptides were tested in HEK293T-cells expressing the Drd2 fused to the N-terminal half of YFP (Drd2-nYFP) and the Drd3 fused to the C-terminal half of YFP (Drd3-cYFP). The heteromer assembly was determined as the YFP-associated fluorescence in bimolecular fluorescence complementation (BiFC) assays in the absence or in the presence of different TM peptides. Among the different peptides derived from TM segments from Drd2, the only peptide able to decrease the YFP fluorescence was the TM5 (Figure 4B), whereas in the case of the Drd3, peptides corresponding to TM2 and TM6 induced a significant interference in the generation of YFP fluorescence (Figure 4C). These results revealed three TM segments able to disrupt the assembly of the Drd2:Drd3 heteromer, suggesting that these segments, TM5 from Drd2, and TM2 and TM6 from Drd3, are involved in the interfaces of interaction required for heteromer assembly.

Afterwards, we speculated whether the Drd2:Drd3 heteromeric complex was expressed in primary colonic Treg cells and, if so, whether the expression of the complex is altered upon inflammation. For this purpose, we treated *Foxp3^gfp^* reporter mice with DSS (1.75%) or only water (healthy controls) for 8 days and then, the presence of the Drd2:Drd3 complex was analysed on Treg (GFP^+^ cells) and non-Treg lymphoid cells (GFP^−^) in the colonic tissue by in situ proximity ligation assay (PLA). Of note, the in situ PLA has been instrumental to determine whether two proteins are in close proximity in primary cells and tissues [34]. The density of Drd2:Drd3 complex expression was quantified as the number of red dots per cell (ratio, R). The results show that, under homeostatic conditions, Treg cells (GFP^+^ cells) express a higher density of Drd2:Drd3 complex expression than non-Treg lymphoid cells (GFP^−^) (Figure 5). Similarly, under inflammatory conditions, the Drd2:Drd3 complex expression was higher in GFP^+^ cells than in GFP^−^ cells (Figure 5). Interestingly, the density of Drd2:Drd3 complex expression increased in both Treg and non-Treg lymphoid cells upon DSS-induced inflammation (Figure 5).

To confirm that the Drd2:Drd3 complex expression detected using PLA was dependent on Drd2 and Drd3, we conducted similar experiments in *Drd2^f/f^*/*CD4^Cre^* (*Drd2*^−/−^), *Drd3*^−/−^ (*Drd3*^−/−^) or *Drd2^+/+^**Drd3^+/+^* (WT) *Foxp3^gfp^* reporter mice treated with DSS or normal drinking water (control) for 8 days followed by in situ PLA analysis in the colonic mucosa. Interestingly, these results revealed that the extent of Treg (GFP^+^ cells) in the colonic mucosa increased with inflammation, an effect that was dependent on Drd2 and Drd3 (Figure 6A,B). The Drd2:Drd3 complex expression on Treg (GFP^+^ cells) was lower in knockout mice both in the density of expression (R; Figure 6C) and in the percentage of PLA^+^ cells (Figure 6D). Also, the density of Drd2:Drd3 complex expression on Treg (GFP^+^ cells) was increased upon DSS treatment only in WT mice, but not in knockout mice (Figure 6C). However, the percentage of PLA^+^ Treg cells was not increased with inflammation in any of the genotypes (Figure 6D). These results indicate that, under inflammatory conditions, the number of Drd2:Drd3 complexes in colonic Treg, but not the number of Treg cells expressing the Drd2:Drd3 complex, increases.

Regarding non-Treg lymphoid cells (GFP^−^), the percentage of cells expressing Drd2:Drd3 complexes was significantly decreased in wild-type (WT) mice upon inflammation (Figure 6D), and these PLA^+^ GFP^−^ cells were barely detectable in DSS-treated or non-treated knockout mice (Figure 6D). The increased density of Drd2:Drd3 complex expression on GFP^−^ cells observed in WT upon inflammation was abrogated only by *Drd3* deficiency, but not in *Drd2^f/f^*/*CD4^Cre^*
*Foxp3^gfp^* mice (Figure 6C). This result might be due to the expression of the Drd2:Drd3 heteromer in another type of lymphoid cells, possibly γδT-cells or innate lymphoid cells.

### 2.5. The Drd2:Drd3 Heteromer Controls the Suppressive Activity and Colonic Tropism of Treg

Since the highest expression of the Drd2:Drd3 heteromer in the colonic mucosa was observed in Treg cells, which further increases upon inflammation, and both Drd2 and Drd3 exert an important regulatory effect on the suppressive activity and colonic tropism of Treg in inflammatory conditions, we next addressed the relevance of the Drd2:Drd3 heteromer in the suppressive activity and colonic tropism of Treg in gut inflammation. Accordingly, we isolated Treg (GFP^+^) from *Foxp3^gfp^* reporter mice which were treated with a peptide able to disrupt the assembly of the Drd2:Drd3 heteromer, the TM5 from Drd2 (TM5Drd2), or with a peptide irrelevant to heteromer assembly (TM1Drd2), and then we determined the suppressive activity in vitro both in the presence or in the absence of dopamine. The results show that the presence of dopamine abrogated the suppressive activity of Treg cells but only when the heteromer was disrupted by TM5Drd2 (Figure 7A,B). The presence of dopamine did not significantly affect the Treg suppressive activity in the absence of peptides or in the presence of the non-disrupting peptide TM1Drd2 (Figure 7A,B). Accordingly, the treatment of Treg with the disrupting peptide TM5Drd2 induced a substantial reduction in the levels of IL-10 in the culture supernatant (Figure 7C).

A further analysis of the molecules involved in the suppressive Treg activity revealed that the Drd2:Drd3 heteromer disruption selectively reduced the expression of CD25, with no effect on the expression of CTLA-4, PD1 and Blimp-1 (Appendix A). Altogether, these results suggest that, in the presence of high dopamine levels, similar to those in the colon under homeostatic conditions (≈2 µM), the intact Drd2:Drd3 heteromer avoids the Drd3 signalling, which inhibits the suppressive activity of Treg [31]. These results also suggest that the Drd2:Drd3 heteromer effect on Treg suppressive activity is mediated by affecting IL-10 production and surface CD25 expression.

Analogously, we next aimed to determine the relevance of Drd2:Drd3 in the ability of Treg to reach the colonic mucosa upon inflammation using in vitro migration assays. To this end, Treg (GFP^+^) isolated from congenic *Foxp3^gfp^* reporter mice were treated with a disrupting peptide (TM5Drd2; CD45.1^+^ CD45.2^+^ cells) or with a peptide irrelevant to heteromer assembly (TM1Drd2; CD45.1^+^ cells) and were i.v. transferred into congenic recipients (CD45.2^+^) previously exposed to DSS treatment for 5 days. The arrival of donor cells to different tissues was analysed 24 h later (Figure 8A). Before the adoptive transfer, we confirmed that the TM5Drd2 and TM1Drd2 peptides did not affect the surface expression of the gut-homing molecules CCR9 and α4β7 integrin on the Treg surface in the absence of dopamine (Figure 8B). Importantly, we observed a substantial reduction in the arrival of TM5Drd2-treated Treg into the colonic lamina propria compared with TM1Drd2-treated Treg (Figure 8C). Nevertheless, the infiltration of TM5Drd2-treated Treg and TM1Drd2-treated Treg in the MLNs was equivalent (Figure 8C) indicating that the Drd2:Drd3 disruption abrogated the recruitment of Treg into the colonic mucosa upon inflammatory conditions. Altogether, these results reveal that the Drd2:Drd3 heteromer expressed on Treg plays an important role, favouring suppressive Treg activity and promoting colonic tropism upon gut inflammation.

## 3. Discussion

Treg cells play a fundamental role in maintaining intestinal homoeostasis and immune tolerance in steady-state conditions. Interestingly, our present results and previous works [35,36,37] show that colonic Treg cells increase in number upon gut inflammation. This change may be related with the active recruitment of Treg into inflamed sites of the intestine, attempting to suppress inflammation. As the balance between Teff and Treg is pivotal in the pathogenesis and progression of IBD, an increased Treg number suggests a compensatory mechanism to counteract a heightened inflammatory response. Accordingly, gut inflammation induces accelerated Treg turnover, proliferation in the colon and bidirectional movement between the colon and the distal part of the MLNs [35]. Another possibility suggests that Treg cells become dysfunctional or even adopt a pro-inflammatory profile in response to signals present in the local microenvironment under inflammatory conditions [3,38]. In this regard, gut inflammation has been associated with a marked reduction in the levels of intestinal dopamine in both human IBD patients and mouse models [10,11,16]. Of note, high dopamine levels and the consequent Drd2 stimulation are associated with anti-inflammatory effects [3,18,39], whereas the selective stimulation of Drd3 in T-cells, which is favoured in the presence of low dopamine levels, promote Th1- and Th17-mediated inflammation [14,15,31,40].

In a previous study conducted in mouse models of gut inflammation, we showed that the selective stimulation of Drd3 in Treg attenuates their suppressive activity and limits their recruitment into the colonic mucosa [31]. Thus, those findings suggest that the decrease in colonic dopamine levels represents a tissue perturbation triggering inflammation. Since high dopamine levels seem to play a homoeostatic role in dopaminergic tissues, we attempted to analyse how the loss of Drd2 signalling in Treg affects their function in the gut. The present data demonstrate that Drd2 signalling promotes Treg function in vivo and attenuates gut inflammation. In fact, a selective Drd2 agonist increased the suppressive Treg activity in vitro**.** Since Drd3 stimulation inhibits the suppressive Treg activity through down-regulating IL-10 production [31], we questioned whether this immunoregulatory cytokine was involved in the mechanism underlying the improved Treg activity triggered by Drd2. Indeed, we found that Drd2 signalling induces greater IL-10 production, which represents one of the main mechanisms by which Tregs limit gut inflammation. In fact, the lack of IL-10 production by Treg cells induces inflammation in the colon [7] and *il10*-deficient mice spontaneously develop colitis [41,42].

In addition to the effects on the suppressive activity, Drd3-mediated signalling dampens the recruitment of Treg into the colonic lamina propria by reducing the surface CCR9 expression [31]. Here we found again that, contrary to the effects promoted by Drd3, Drd2 signalling increases the CCR9 expression on Treg cells, improving their infiltration into the colonic mucosa. Since our observations indicated that dopamine, depending on the concentration, might trigger opposite biological effects on Treg cells, we hypothesised that Drd2 and Drd3 form a heteromeric receptor in this lymphocyte population. Evidence has shown that GPCR heteromers constitute a functional unit of physiological relevance, which trigger different biological effects than those triggered by the isolated forms of the receptors that conform to the heteromer [43]. An interesting example of a GPCR heteromer that triggers opposite effects depending on the concentration of the same ligand is the complex formed by adenosine receptors A_2A_ (A_2A_R) and A_1_R in striatal glutamatergic terminals. Under basal conditions, low levels of adenosine preferentially stimulate A_1_R, which displays a higher affinity for adenosine than A_2A_R, inhibiting glutamatergic transmission. However, upon high adenosine concentrations, the stimulation of A_2A_R in the A_2A_R:A_1_R heteromer blocks A_1_R signalling, thus stimulating glutamate release from striatal nerve terminals [44].

The ability of Drd2 and Drd3 to assemble into a functional Drd2:Drd3 heteromer was previously demonstrated when co-expressed in a heterologous system [45,46]. Co-immunoprecipitation [45] and experiments using SNAP and CLIP tag covalent labelling reagents followed by time-resolved fluorescence resonance energy transfer [46] demonstrate that Drd2 and Drd3 form both homomers and heteromers in steady-state conditions [46]. In the present study, the BRET analysis of HEK293T-cells transfected with RLuc fused to Drd3 and YFP fused to Drd2 confirmed the physical interaction between these two receptors. The specificity of this interaction was confirmed as YFP-Drd2 did not interact with RLuc-A_1_R, since the co-transfection of these constructs led to a barely detectable BRET signal. To further explore the molecular requirements involved in the assembly of the Drd2:Drd3 heteromer, we studied the ability of α-helix peptides analogous to the TM segments from Drd2 and Drd3 to disrupt the assembly of the Drd2:Drd3 heteromer using a BiFC approach. Our results revealed that only TM5 from Drd2 and TM2 and TM6 from Drd3 were able to disrupt the heteromer assembly, indicating that only these three TM segments are involved in the interacting interface.

Importantly, our PLA analysis revealed for the first time that the Drd2:Drd3 heteromeric complex is expressed in primary lymphocytes, particularly in colonic Treg cells. Interestingly, the expression density in colonic Treg increased under inflammatory conditions, suggesting a functional relevance in this pathophysiological context. As part of the physiological response of immune cells to inflammation induced by DSS treatment, a critical step involves the formation of actin-based structures required for migration into the gut. In this regard, the heteromerization of Drd2 and Drd3 might be influenced by proteins that regulate their direct interaction, trafficking and desensitisation [47]. For instance, these two dopamine receptors are linked to the actin cytoskeleton through the interaction with filamin-A and mutations in the cytoskeletal-binding domain and the region of the dopamine receptor binding of filamin-A reduces the cell surface expression of Drd2 and Drd3 [48,49]. It seems therefore plausible that the surface expression of Drd2:Drd3 heteromers and Treg migration during gut inflammation are linked. Consistent with this, our results indicate that the increased accumulation of colonic Treg during gut inflammation is dependent on both Drd2 and Drd3 expression and correlates with an enhanced surface expression of the heteromer on colonic Treg under inflammatory conditions. Of note, the reduced Treg infiltration in the colonic mucosa of *Drd3*-deficient mice contradicts our previous study showing that *Drd3* deficiency in Treg improves their recruitment into the colonic lamina propria upon inflammation, where the recruitment of *Drd3*-sufficient and *Drd3*-deficient Treg transferred into DSS-treated wild-type recipients was compared [31]. Hence, the different results observed here on the extent of Treg arrival to the colon may be attributed to Drd3 deficiency in other cell types influencing Treg infiltration into the colonic mucosa.

Addressing the functional relevance of the Drd2:Drd3 heteromeric complex in Treg cells, Drd2:Drd3 heteromer signalling is largely responsible for Treg suppressive activity in vitro as the disruption of the heteromer abrogated their suppressive function, with dopamine levels similar to those in the colon under homeostatic conditions. These findings highlight the physiological relevance of the Drd2:Drd3 heteromer in maintaining immune homeostasis in the colonic mucosa. Furthermore, the Drd2:Drd3 heteromer was also responsible for the proper recruitment of Treg cells into the colonic mucosa under inflammation, as disrupting the heteromer assembly strongly reduced the arrival of Treg in the colonic lamina propria of DSS-treated mice. This indicates that the integrity of the heteromer is essential for the efficient recruitment of Treg into inflamed intestines. Thereby, any event that limits or interferes with the formation of this heteromer might worsen inflammation, as Treg are essential for maintaining immune tolerance and controlling excessive immune responses. Since the stimulation of Drd2 and Drd3 triggers opposite functions in Treg cells, including gut tropism and suppressive activity, we aimed to identify signalling pathways coupled to Drd2 and Drd3 in opposite directions. However, we found that cAMP, ERK1/2 phosphorylation and AKT phosphorylation were all coupled to Drd2 and Drd3 stimulation in the same direction (Appendix A). Since CD25 expression was significantly affected by the Drd2:Drd3 heteromer disruption (Appendix A), and Blimp-1 negatively regulates CD25 expression on Treg [50], we also evaluated how Blimp-1 expression is affected by the heteromer integrity; however we did not find significant effects (Appendix A). Future research should aim to understand the molecular mechanisms by which Drd2 and Drd3 stimulation in the heteromer trigger opposite functional effects at the level of migration to the gut and suppressive activity.

Dopaminergic signalling plays a broader role in regulating leukocyte migration, as a heteromeric receptor formed by CCR9 and Drd5. In response to elevated CCL25 and reduced dopamine levels, it promoted effector CD4^+^ T-cell recruitment into the colonic lamina propria during gut inflammation [29]. The dual stimulation of this heteromer triggers the phosphorylation of the myosin light chain 2 in CD4^+^ T-cells, enhancing their motility and migratory speed, which in turn improves their infiltration into the inflamed colonic mucosa [30]. Another example, already commented here, is the regulation of Treg migration by DRD3-mediated signalling, which limits the recruitment of these cells into the colonic mucosa by down-regulating CCR9 expression [31]. In addition, Drd3 signalling in naive CD8^+^ T-cells favours the recruitment of these cells into the lymph nodes by potentiating the migration towards CCL19 and CCL21 [51]. The ability of dopamine for regulating migration is not limited to lymphoid cells but also occurs in myeloid cells. Indeed, DRD4 stimulation in macrophages induces an up-regulation of CCR5, increasing their migratory ability to infiltrate the brain [52]. Interestingly, the same laboratory reported that DRD1-mediated signalling triggered the opposite effect, down-regulating CCR5 and thereby impairing the macrophage ability to infiltrate the brain [52], illustrating again how dopamine, depending on high or low concentrations, might exert opposite biological effects on leukocytes.

Dopamine receptor heteromerization represents a new dimension for interpreting the actions of this neurotransmitter in physiological and pathological conditions, thus opening new opportunities for the design of more specific therapies. In the context of IBD, a number of therapies have involved the development of antibodies or small molecules designed to interfere with CCR9-CCL25 or α4β7-MadCAM1 interactions to prevent the recruitment of T-cells into the inflamed gut mucosa [53,54]. However, these interactions are also essential for the recruitment of Treg cells into the intestinal mucosa during homeostasis, which is crucial for maintaining immune tolerance [27]. As a result, blocking CCR9-CCL25 and α4β7-MadCAM1 interactions may lead to serious collateral effects. The emerging understanding of the more intricate control of leukocyte migration, involving specific GPCR heteromers expressed in distinct leukocyte subsets, offers the potential for developing small molecules or monoclonal antibodies that selectively target immune cell recruitment in inflamed tissues. Consequently, a deeper insight into GPCR heteromers regulating leukocyte biology could drive the creation of next-generation therapeutic agents, capable of providing more precise benefits while minimising side effects in the treatment of IBD and other inflammatory disorders.

## 4. Materials and Methods

### 4.1. Animals

Wild-type (*CD45.1*^+/+^; *CD45.2*^+/+^), *CD4^Cre^* and *Rag1*^−/−^ mice were obtained from The Jackson Laboratory. *Foxp3^gfp^*
*CD45.1*^+/+^ reporter mice generated as described before [55] were also obtained from The Jackson Laboratory. *Drd2^flox/flox^* (*Drd2^f/f^*) mice were kindly donated by Dr. Jiawei Zhou [18], and *Drd3*^−/−^ mice were kindly donated by Dr. Marc Caron [56]. *CD4^Cre^*/*Drd2^f/f^*, *CD4^Cre^*/*Drd2^f/f^*/*Foxp3^gfp^*, *Drd3*^−/−^/*Foxp3^gfp^* and *Cd45.1*^+/−^/*Cd45.2*^+/−^ mice were generated by crossing parental mouse strains. We confirmed the genotype of these new strains by PCR of genomic DNA and by flow cytometry of peripheral blood cells (for GFP). Female mice from 6 to 10 wk were used in all experiments. All mice were housed in plastic home cages in a temperature-controlled room, under a 12:12 h light cycle. The environment was enriched with sterile cardboard cones. All animals had ad libitum access to standard rodent food pellets and drinking water. Experimental mice were monitored daily and received a health score (from 0 to 12) based on body weight loss (from 0 to 3), general healthy aspect (from 0 to 3), spontaneous behaviour (from 0 to 3) and stool consistency (from 0 to 3). When the health score was ≥7, mice were euthanised with CO_2_ overdose. When the health score was lower than 7 and difficulty eating or drinking was noticed, food and hydration were provided in the form of wet jellies. All procedures performed on animals were approved by and complied with regulations of the Institutional Animal Care and Use Committee at Fundación Ciencia & Vida; approved on 31 July 2020 (permit number: P016-2020).

### 4.2. Reagents

Peptides analogous to the TM segments of Drd2 and Drd3 (Table 1) were synthesised by GenScript (Hong Kong). Anti-CCR9-PE-Cy7 monoclonal antibody (mAb) was obtained from eBioscience (San Diego, CA, USA). Anti-α4β7-PE, anti-CD25-FitC, anti-CD4-PercP, anti-CD4-APC, anti-CD45.1-BV421, anti-CD45.2-Pe-Cy7, anti-Blimp-1-PE, anti-PD-1-APC and anti-CTLA-4-BV421 mAbs and ZAq Fixable Viability kit were purchased from Biolegend (San Francisco, CA, USA). For intracellular cytokine staining, restimulation was performed with phorbol 12-myristate 13-acetate (PMA) and ionomycin (both from Sigma-Aldrich; St. Louis, MO, USA), and brefeldin A (from Life Technologies; Waltham, MA, USA), fixation/permeabilization solution and the permeabilization buffer were obtained from eBioscience (San Diego, CA, USA). For the induction of intestinal tropism, Treg were treated with ultra-purified LEAF anti-mouse CD3ε (BioLegend, San Francisco, CA, USA), anti-mouse CD28 (TONBO Biosciences, San Diego, CA, USA), all-trans retinoic acid (Sigma-Aldrich, St. Louis, MO, USA), recombinant mouse IL-2 protein (PeproTech, Rocky Hill, NJ, USA), anti-IFNγ (BioLegend, San Francisco, CA, USA) and TGF-β1 (Sigma-Aldrich, St. Louis, MO, USA). For the in vitro migration assay, fibronectin (Sigma-Aldrich; St. Louis, MO, USA), recombinant CCL25 protein (Biolegend; San Francisco, CA, USA) and 123 count eBeads (eBioscience, San Diego, CA) were used. Forskolin and the dopaminergic analogues, sumanirole maleate and 7-Hydroxy-PIPAT maleate, were obtained from Tocris (Bristol, UK). The DSS was purchased from Tdb Labs (Uppsala, Sweden). All reagents related to the culture medium were obtained from Life Technologies (Waltham, MA, USA). For PLA experiments, Coelenterazine H (Molecular Probes; Eugene, OR, USA), Goat anti-Drd2 antibody (cat# sc7522, Santa Cruz Biotechnology; Dallas, TX, USA) and rabbit anti-Drd3 antibody (cat# ab42114; Abcam; Cambridge, UK) were used. Duolink^®^ In Situ PLA Detection Kit, Duolink^®^ In Situ PLA^®^ Probe Anti-Goat PLUS, Duolink^®^ In Situ PLA^®^ Probe Anti-Rabbit MINUS and and Hoechst were obtained from Merck/Sigma-Aldrich (Uppsala, Sweden). For the determination of AKT/ERK phosphorylation, protease inhibitor mixture and PVDF membranes (Immobilon-FLPVDF membrane) were obtained from Merck (St. Louis, MO, USA); the bicinchoninic acid method for protein quantification and BSA were purchased from ThermoFisher Scientific (Waltham, MA, USA); Odyssey Blocking Buffer was obtained from LI-COR Biosciences (Lincoln, NE, USA); rabbit anti-ERK1/2 (Ref. M5670) and IRDye 680 anti-rabbit antibody (Ref. 926-68071) were obtained from Merck (St. Louis, MO, USA); and rabbit anti-phospho-AKT antibody (Ref. 11054) was purchased from Signalway Antibody (Baltimore, MA, USA).

### 4.3. Dextran Sodium Sulphate Induced Acute Inflammatory Colitis

Experimental groups were made by selecting female mice randomly and paired by age. Mice displaying dwarfism or malformations were excluded a priori. Mouse groups with different treatments and/or genotypes were kept in different cages to avoid confounding factors. For blinding evaluation, JM or VU was the investigator who knew the identity of the evaluated mice. Mice were treated with 1% (for suboptimal conditions) or 1.75% (optimal conditions) DSS in the drinking water. DSS was given for a total period of 7 d and then replaced with normal drinking water until the end of the experiment. Body weight was recorded throughout the time course of disease development. The extent of the loss of initial body weight was used as the main parameter to determine disease severity. At the end of the experiment, mice were sacrificed to obtain the spleen, MLNs and cLP. Tissue was digested and homogenised using gentleMACS^TM^ dissociator (Miltenyi Biotec; Bergisch Gladbach, Germany) and then filtered through cell strainers (70 μm pore). In some cases, the colon was fixed and used for in situ PLA, while in other cases used to obtain mononuclear cells (MNCs) from cLP. For the latter purpose, cells were separated using centrifugation in percoll [57]. MLN cells were restimulated with PMA and ionomycin in the presence of brefeldin A and intracellular IL-10 production by T-cells was analysed by flow cytometry. The outcome measurement was the percentage of body weight loss respective to the initial body weight.

### 4.4. Histological Analysis

Mice were sacrificed and the colons were excised. A representative piece of distal colon from each mouse was fixed in 10% formaldehyde and processed for staining with hematoxylin and eosin (H&E) staining. A blinded histopathologic evaluation of colons was performed considering inflammation, the extent of injury, crypt damage and the percentage of tissue involved, as indicated in Table 2, by Merken Biotech. RP was the only person who knows the identity of mice evaluated. The outcome measurement was the histopathological score of colonic tissue.

### 4.5. In Vitro Suppression Assay

Treg (CD4^+^ GFP^+^) obtained from *CD45.1*^+/+^
*Foxp3^gfp^* reporter mice were incubated in 200 μL RPMI medium containing 100 ng retinoic acid (RA) and 200 IU of IL-2, and activated with 60 ng of plate-bound anti-CD3 and 100 ng of soluble anti-CD28 for 4 h. While activating, Treg were incubated with the indicated peptides (Table 1; GenScript) during 4 h, or with the indicated drugs during the last 30 min, and then washed. Naive CD4^+^ CD25^−^ T-cells (T naive) isolated from WT *CD45.2*^+/+^ mice were loaded with 5 μM cell trace violet (CTV) and co-cultured (10^5^ cells/well) with activating Treg at the indicated Treg/T naive ratios in 96-well plates in the presence of anti-CD3 and anti-CD28 Abs. After 3 d, the extent of T naive proliferation was determined as the dilution of CTV-associated fluorescence in the CD4^+^ GFP^−^ ZAq^−^ population by flow cytometry. CD25, CTLA-4, PD1 and Blimp-1 expression were analysed in Treg cells by flow cytometry, and the levels of IL-10 in the supernatant were quantified by ELISA. The outcome measurement was the percentage of proliferating naive T-cells, or the percentage of the expression of molecular markers in Treg (CD25, CTLA-4, PD1 and Blimp-1) or the concentration of IL-10 in the culture supernatant in response to the different treatments.

### 4.6. In Vivo Migration Assay

Splenic Treg cells (CD4^+^GFP^+^) isolated from *CD45.1*^+/+^
*Foxp3^gfp^*, *CD45.1*^+/−^
*CD45.2^+/−^*
*Foxp3^gfp^* or *CD45.2*^+/+^^−^
*Foxp3^gfp^*
*Drd2^f/f^*
*CD4^Cre^* mice were incubated with or without the indicated peptides (Table 1; 4 μM) and activated for 4 h, as indicated above (see Section 4.5). Then, donor CD4^+^ T-cells were mixed in a 1:1 ratio and 7 × 10^5^ total cells were i.v. injected into *CD45.1*^+/−^, *CD45.2*^+/−^ or *CD45.2*^+/+^ *Rag1*^−/−^ recipient mice exposed to DSS 1.75%. Mice were sacrificed 24 h (when recipients were *Rag1*^−/−^) or 48 h (when recipients were WT) later and the relative composition (CD45.1^+^ versus CD45.2^+^) on CD4^+^ T-cells isolated from different tissues were analysed, including the spleen, MLNs and cLP. The quantification of the relative abundance of CD45.1^+^ versus CD45.2^+^ CD4^+^ T-cells was normalised with the input composition. The outcome measurement was the ratio of the output/input calculated with the percentage of donor cells from different experimental groups present in the tissue assessed and in the initial mixture (previous to the adoptive transfer).

### 4.7. In Vitro Migration Assay

Naive T-cells (CD4^+^CD62L^+^CD44^−^GFP^−^) isolated from the MLNs of *Drd2^f/f^*/*CD4^Cre^*
*Foxp3^gfp^* or *Drd2*^+/+^
*Foxp3^gfp^* mice were incubated in 200 μL RPMI medium containing 100 ng RA, 20 ng TGF-β1, 100 ng anti-IFNγ Ab and 200 IU of IL-2, and activated with 60 ng of plate-bound anti-CD3 and 100 ng of soluble anti-CD28 for 5 d to induce the differentiation into iTreg. RA, TGF-β1, anti-IFNγ Ab and IL-2 were renewed at days 2 and 4. 3 × 10^5^ live iTreg cells were resuspended in 100 µL (PBS) and seeded on the top chamber of 5 µm pore transwells (Corning, NY, USA). Furthermore, 2 h before, the bottom chamber was incubated with fibronectin (10 µg/mL; Sigma Aldrich) in 600 µL of RPMI containing 5% BSA and either mouse CCL25 (300 ng/mL), PBS or mouse serum. Cells were incubated at 37 °C and 5% CO_2_ for 4 h. Then, both top and bottom chamber cells were recovered, stained with ZAq for 15 min and resuspended in 150 µL of PBS. To quantify the absolute number of cells, 5 µL of 123 count eBeads was added to each sample prior to flow cytometry analysis and cell concentration was calculated as indicated by manufacturer’s instructions. The outcome measurement was the number of Treg cells reaching the bottom chamber of the transwell.

### 4.8. Flow Cytometry Analysis

Cells were stained with a ZAq Fixable Viability kit, followed by fluorochrome-conjugated mAbs specific to cell surface markers in PBS containing 5% FBS for 15 min. Surface markers analysed included α4β7, CCR9, CD4, CD25, CD45.1, CD45.2, CD44, CD62L and TCRβ. Afterwards, cells were fixed with 1% paraformaldehyde in phosphate-buffered saline (PBS, Na_2_HPO_4_ 8.1 µM, KH_2_PO_4_ 1.47 µM, NaCl 64.2 mM, KCl 2.68 mM, pH 7.4) for 15 min at room temperature, washed twice with PBS and analysed in a flow cytometer. For intracellular cytokine staining (IL-10), CD4^+^ T-cells were stimulated for 4 h with phorbol-12-myristate-13-acetate (PMA, 50 ng mL^−1^) and ionomycin (1 μg mL^−1^) in the presence of brefeldin A (5 μg mL^−1^, Life Technologies). Cells were stained with a ZAq Fixable Viability kit, followed by cell surface marker immunostaining in PBS containing 2% FBS. Afterwards, cells were resuspended in fixation/permeabilization solution and incubated for, at least, 30 min. Then, intracellular immunostaining was carried out in permeabilization buffer at 4 °C for 1 h. Data were collected with a Canto II (BD) and results were analysed with FACSDiva (BD) and FlowJo software (New V9/10; Tree Star, Ashlan, OR, USA). All analyses were controlled at the first step with autofluorescence and at the second step with Fluorescence Minus One (FMO).

### 4.9. Bulk RNA-Seq Analysis

The raw RNA-seq data underwent quality assessment using FastQC (v0.11.7) with default parameters. Adapters identified by FastQC were removed using Cutadapt (v1.1) with default settings. Sickle (v1.200) was then employed to trim low-quality ends of reads, retaining sequences with a minimum length of 25 nucleotides and a quality score threshold of 20. The cleaned reads were aligned to the human genome (GRCh37, human_glk_v37) using HISAT (v0.1.6), allowing for a maximum of two mismatches. Aligned reads were sorted using SAMtools (v0.1.19), and mapping statistics were generated using SAMtools flagstat and Picard tools (v2.9.0). Samples with a low percentage of aligned reads (<90%) were excluded. Gene-level expression was quantified using HTSeq (v0.9.1) with annotation from Ensembl version 75. Data normalisation was performed using the trimmed mean of M-values (TMM) method to account for differences in library sizes. The outcome measurement was the normalised amount of DRD2 and DRD3 transcripts in the biopsies.

### 4.10. Bioluminescence Resonance Energy Transfer Assay

For bioluminescence resonance energy transfer (BRET) experiments, HEK293T-cells transiently co-transfected with a constant amount of cDNA encoding *Drd3* (or *A1R* as a control) fused to RLuc and with increasing amounts of cDNA encoding *Drd2* fused to YFP (see figure legends) were used 48 h after transfection. To quantify BRET measurements, 5 μM coelenterazine H (Molecular Probes, Eugene, OR, USA) was added to the equivalent of 20 μg of cell suspension. After 1 min, the readings were collected using a Mithras LB 940 (Berthold Technologies, Bad Wildbad, Germany) that allows the integration of the signals detected in the short-wavelength filter at 485 nm and the long-wavelength filter at 530 nm. To quantify protein-RLuc expression, luminescence readings were also performed after 10 min of adding 5 μM coelenterazine H. To quantify protein-YFP expression, the fluorescence of cells (20 μg protein) was also read. The net BRET is defined as [(long-wavelength emission)/(short-wavelength emission)]-Cf where Cf corresponds to [(long-wavelength emission)/(short-wavelength emission)] for the donor construct expressed alone in the same experiment. Data were fitted to a non-linear regression equation, assuming a single-phase saturation curve with GraphPad Prism 10 software (San Diego, CA, USA). BRET is expressed as milli BRET units, mBU (net BRET × 1000). The outcome measurement was the relative bioluminescence associated with the oxidisation of luciferin in the presence of different amounts of the fusion proteins transfected.

### 4.11. Bimolecular Fluorescence Complementation Assay

HEK293T-cells were transiently transfected with equal amounts of the cDNA for fusion proteins of the hemi-truncated Venus (1.5 µg of each cDNA). Next, 48 h after transfection, cells were treated for 4 h at 37 °C with TM-analogue peptides (0.4 µM) before plating 20 μg of protein in 96-well black microplates (Porvair, King’s lynn, UK). To control the cell number, the sample protein concentration was determined by a Bradford assay kit (Bio-Rad, Munich, Germany) using bovine serum albumin (BSA) dilutions as standard. To quantify fluorescent proteins, cells (20 g of total protein) were distributed in 96-well microplates (black plates with a transparent bottom) and fluorescence was read in a Fluostar Optima Fluorimeter (BMG Labtech, Ofenburg, Germany) equipped with a high-energy xenon flash lamp using a 10 nm bandwidth excitation filter at 485 nm. Protein fluorescence expression was determined as the fluorescence of the sample minus the fluorescence of cells not expressing the fusion proteins (basal). The outcome measurement was the relative fluorescence associated with the assembled Venus fluorescent protein in the presence of different treatments.

### 4.12. In Situ Proximity Ligation Assay

Colonic sections of healthy or DSS-treated mice were used to analyse the Drd2:Drd3 heteromer in situ by proximity ligation assay (PLA). Tissue sections were fixed in 4% paraformaldehyde for 15 min, washed with PBS containing 20 mM glycine to quench the aldehyde groups and permeabilized with the same buffer containing 0.05% Triton X-100 for 30 min. Primary antibodies recognising Drd2 (rabbit anti-Drd2; 1:100 dilution) and Drd3 (rabbit anti-Drd3; 1:100 dilution) were used. Primary antibodies were linked directly to PLA probes detecting rabbit antibodies (Duolink II PLA probe anti-goat plus and Duolink II PLA probe anti-rabbit minus). As negative technical controls, samples followed the same procedure but in the absence of anti-Drd2 primary antibodies. After 1 h incubation at 37 °C with blocking solution, tissue sections were incubated with the primary antibodies linked to PLA probes and further processed as described before [58]. Nuclei were stained with Hoechst (1:200 dilution). Coverslips were mounted using mowiol solution. Samples were observed in a Leica SP2 confocal microscope (Leica Microsystems, Mannheim, Germany) equipped with an apochromatic 63X oil immersion objective (N.A. 1.4), and 405 nm and 561 nm laser lines. For each field of view, a stack of two channels (one per staining) and 3 to 4 Z stacks with a step size of 1 µm were acquired. The quantification of cells containing one or more red spots versus total cells (blue nucleus) and, in cells containing spots, the ratio r (number of red spots/cell) was determined by Andy’s algorithms [59]. The outcome measurement was the percentage of lymphoid cells expressing Drd2 and Drd3 in proximity, and the number of Drd2:Drd3 complexes detected per cell (R).

### 4.13. cAMP Levels Determination

HEK293T-cells expressing DRD2 and DRD3 were incubated in serum-free medium for 4 h. Cells were plated in 384-well white microplates (1000 cells/well) and incubated for 15 min with the specific agonists and antagonists followed by 15 min stimulation with 0.5 µM forskolin. cAMP production was quantified by a TR-FRET (Time-Resolved Fluorescence Resonance Energy Transfer) methodology using the LANCE Ultra cAMP kit (PerkinElmer) and the Pherastar Flagship Microplate Reader (BMG Labtech, Ortenberg, Germany). The outcome measurement was the percentage of cAMP production in response to treatments.

### 4.14. Determination of ERK1/2 Phosphorylation

HEK293T-cells expressing DRD2 and DRD3 were incubated in serum-free medium for 2 h. ERK1/2 phosphorylation was determined using an AlphaScreen^®^SureFire^®^ kit (Perkin Elmer) following the instructions of the supplier and using an EnSpire^®^ Multimode Plate Reader (PerkinElmer, Waltham, MA, USA). Cells (30.000 cells/well for transfected HEK293T-cells) were seeded in white ProxiPlate 384-well microplates, pre-treated at 25 °C for 20 min with vehicle or antagonists in serum-starved DMEM medium supplemented or not with 1 μM ionomycin and stimulated for an additional 7 min with the indicated agonists. Phosphorylation was determined by alpha-screen bead-based technology using the Amplified Luminiscent Proximity Homogeneous Assay kit (PerkinElmer, Waltham, MA, USA) and the Enspire Multimode Plate Reader (PerkinElmer). The outcome measurement was the percentage of phosphorylated ERK1/2 in response to treatments.

### 4.15. Determination of AKT Phosphorylation

HEK-293T-cells were transfected with the cDNA encoding for *DRD2* (1 μg) and for *DRD3* (1.5 μg). Two to four hours before initiating the experiment, the culture medium was replaced by serum-starved DMEM medium. Cells were stimulated (7 min) with the corresponding agonists at 37 °C. Then, the reaction was stopped by placing cells on ice. Then, cells were washed twice with cold PBS and lysed by the addition of ice-cold lysis buffer (50 mM Tris-HCl pH 7.4, 50 mM NaF, 150 mM NaCl, 45 mM ß-glycerolphosphate, 1% Triton X-100, 20 mM _Mphenyl-arsine oxide, 0.4 mM NaVO4 and protease inhibitor mixture). Cellular debris were removed by centrifugation at 12.000 rpm for 10 min at 4 °C and protein was adjusted to 1 mg/mL after quantification by the bicinchoninic acid method using BSA for standardisation. Finally, proteins were denatured by placing them at 100 °C for 5 min. AKT phosphorylation was determined by Western blot. Equivalent amounts of protein (20 µg) were subjected to electrophoresis (10% SDS-polyacrylamide gel) and transferred onto PVDF membranes (Immobilon-FLPVDF membrane) for 90 min. Then, the membranes were blocked for 1 h at room temperature (constant shaking) with Odyssey Blocking Buffer and labelled with a mixture of primary rabbit anti-ERK 1/2 antibody (1:40,000), and primary rabbit anti-phospho-AKT antibody (1:2500) overnight at 4 °C with shaking. Then, membranes were washed three times with PBS containing 0.05% tween and visualised by the addition of IRDye 680 anti-rabbit antibody (1:10,000) for 2 h at room temperature. Membranes were washed 3 times with PBS-tween 0.05% and once with PBS and left to dry. Bands were analysed using an Odyssey infrared scanner (LI-COR Biosciences). Band densities were quantified using the fiji software (version 2.9.0), and the level of phosphorylated AKT was normalised using the total ERK 1/2 protein band intensities. The results obtained are represented as the percentage over basal (non-stimulated cells). The outcome measurement was the percentage of phosphorylated AKT in response to treatments.

### 4.16. Statistical Analyses and Sample Size Estimation

The sample size was estimated using the mean and dispersion obtained from preliminary data using the following sample size calculator: https://www.stat.ubc.ca/~rollin/stats/ssize/n2.html accessed on 15 March 2020. A power of 80% was assumed. No data was excluded from the analysis. The main parameters considered for the sample size used were as follows: for DSS experiments it was the body weight loss; for in vitro suppressive assays it was the percentage of proliferating naive T-cells; and for in vivo migration assays it was the ratio output/input calculated with the percentage of donor cells from different experimental groups present in the colon and in the initial mixture (previous to the adoptive transfer). The normality of data was assessed using the Shapiro–Wilk test. For data with a normal distribution, significant differences were calculated with a two-tailed unpaired Student’s *t*-test when comparing only two groups and with one-way ANOVA when comparing more than two groups with only one variable (treatment or genotype). Two-way ANOVA was used to analyse differences in experiments comparing distinct genotypes and/or treatments. For data displaying a non-normal distribution, Kruskal–Wallis’s test was used to compare more than two experimental groups. All analyses were conducted using the GraphPad Prism 10 Software. *p*-values < 0.05 were considered significant.

## 5. Conclusions

The transcriptomic analysis conducted on the intestinal mucosa from IBD patients revealed an association with increased *DRD3* and reduced *DRD2* transcript expression. Furthermore, we find that Drd2 and Drd3 form a heteromeric complex that works as a dopamine sensor, triggering different biological effects on Treg depending on the levels of dopamine. Our findings indicate that the Drd2:Drd3 heteromer constitutes a dopamine sensor that plays a critical role on suppressive Treg activity and colonic tropism in homeostasis and under inflammation. Moreover, our study suggests that the reduction in dopamine levels associated with gut inflammation and the consequent shift in dopamine receptors stimulated in this heteromeric complex expressed on mucosal Treg cells represents one of the molecular changes responsible for Treg unresponsiveness observed in the gut mucosa of IBD patients (Appendix A).

## Figures and Tables

**Figure 1 ijms-26-10069-f001:**
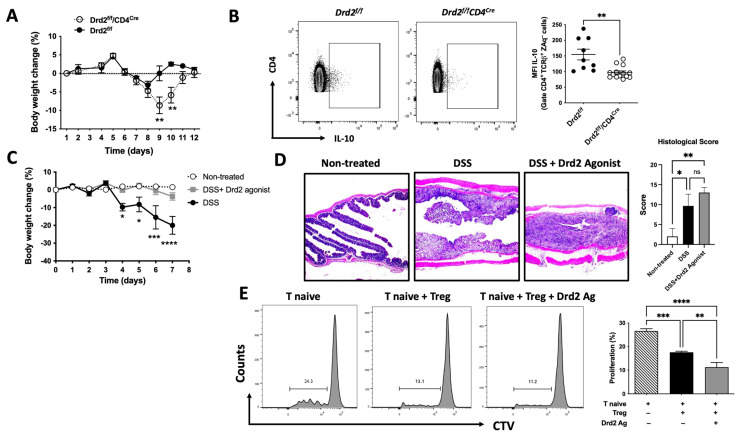
Drd2-mediated signalling promotes Treg function and attenuates gut inflammation. (**A**,**B**) Eight to ten-week-old *Drd2^flox/flox^* (*Drd2^f/f^*) and *Drd2^f/f^*/*CD4^Cre^* mice (*n* = 9–14 mice per group) received a suboptimal dose of DSS (1%) in their drinking water for 7 d. (**A**) The percentage change relative to initial body weight was quantified. Data are the mean ± SEM. (**B**) Mononuclear cells were extracted from the MLNs of *Drd2^f/f^*/*CD4^Cre^* and *Drd2^f/f^* mice and stimulated ex vivo with PMA+Ionomycin in the presence of brefeldin A, and the extent of IL-10 production was analysed in live CD4^+^ T-cells (CD4^+^ TCRβ^+^ ZAq^−^) by intracellular cytokine staining followed by flow cytometry analysis. IL-10 production was quantified as the mean fluorescence intensity (MFI) associated with IL-10 immunostaining. Left panels show representative dot plots. The right panel shows the quantification. Each symbol represents data from an individual mouse. Mean ± SEM is shown. (**C**,**D**) Eight to ten-week-old wild-type mice were non-treated (as a control) or received 1.75% DSS in the drinking water for 7 d. Next, 24 h after the beginning of DSS treatment, a group of animals received a single i.p. injection of a Drd2 agonist (sumanirole; 4 mg/kg), while the other group received the vehicle (control). (**C**) The percentage change relative to initial body weight was quantified. (**D**) Histological analysis. Representative images of H&E staining (left panels) and quantification of histopathological score (right panel). (**C**,**D**) Data are the mean ± SEM from 4 mice per group. (**E**) Naive CD4^+^ CD25^−^ T-cells (T naive) isolated from WT CD45.1^+/+^ mice were loaded with 5 μM CTV and activated with dynabeads coated with anti-CD3 and anti-CD28 Abs and co-cultured with CD4^+^ CD25^+^ GFP^+^ Treg (ratio Treg/Tnaive = 1:8) isolated from CD45.2^+/+^ *Foxp3^gfp^* mice. Before co-culture, Treg cells were pre-incubated with 100 nM sumanirole or vehicle for 30 min. A control group was incubated without Treg. After 72 h, the extent of T naive proliferation was determined as the dilution of CTV-associated fluorescence in the CD4^+^ CD45.1^+^ GFP^−^ ZAq^−^ population by flow cytometry. Left panels show representative dot plots. The right panel shows the quantification. Values are the mean ± SEM of triplicates from a representative experiment. Data from one out of three independent experiments are shown. *, *p* < 0.05; **, *p* < 0.01; ***, *p* < 0.001; ****, *p* < 0.0001 by unpaired Student’s *t*-test (**A**–**C**) or one-way ANOVA followed by Tukey’s post hoc test (**D**,**E**). ns, non-significant.

**Figure 2 ijms-26-10069-f002:**
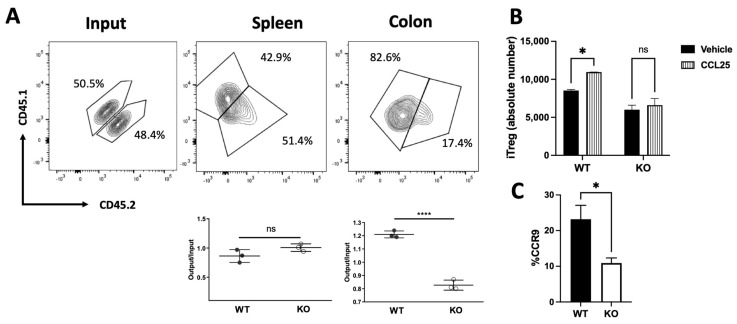
Drd2-mediated signalling promotes Treg recruitment and retention in the colonic mucosa. (**A**) Treg cells (CD4^+^GFP^+^) isolated from the spleen of wild-type (WT; CD45.1^+^; black symbols) or from *Drd2^f/f^*/*CD4^Cre^* (KO; CD45.2^+^; white symbols) *Foxp3^gfp^* mice were mixed in 1:1 ratio (input) and then i.v. injected (7 × 10^5^ total cells per mouse) into WT (CD45.2^+^ CD45.1^+^) recipient mice that previously received DSS for 72 h. Mice were further treated with 1.75% DSS for 48 h after T-cell transfer and then were sacrificed and the relative composition (CD45.1^+^ versus CD45.2^+^) of GFP^+^ Treg isolated from the spleen or colon was analysed. Top panels show representative contour plots of donor Treg in the input or isolated from recipients. Bottom panels show the quantification of the relative abundance of WT or KO Treg in the spleen (left) or colon (right). Data is the % of single positive CD45.1^+^ (WT) or double positive CD45.2^+^ (KO) donor cells in each tissue. Each symbol represents data obtained from an individual mouse. Mean ± SEM are indicated. Data from a representative experiment are shown. (**B**) Naive T-cells (CD4^+^CD62L^+^CD44^−^GFP^−^) isolated from MLNs of *Drd2^f/f^*/*CD4^Cre^ Foxp3^gfp^* (KO) or *Drd2*^+/+^
*Foxp3^gfp^* (WT) mice were differentiated into iTreg for 3 d and then, the migration to CCL25 or vehicle was evaluated by transwell assay. Values are the number of iTreg that arrived into the bottom chamber. Values are mean ± SEM. (**C**) Naive T-cells from *Drd2^f/f^*/*CD4^Cre^ Foxp3^gfp^* (KO) or *Drd2*^+/+^
*Foxp3^gfp^* (WT) mice were differentiated into iTreg and CCR9 expression was determined. Values are the percentage of CCR9^+^ cells in the GFP^+^ ZAq^−^ population. Mean ± SEM from 4 mice per group. *, *p* < 0.05; ****, *p* < 0.0001 by unpaired *t*-test (**A**,**C**) or two-way ANOVA followed by Sidak’s post hoc test (**B**). ns, non-significant.

**Figure 3 ijms-26-10069-f003:**
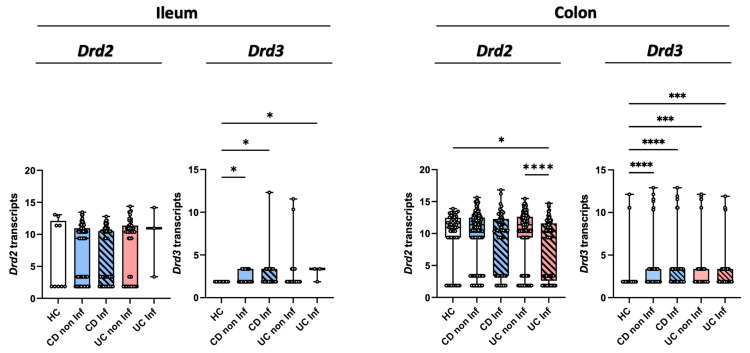
Reduced DRD2 and increased DRD3 expression in inflamed intestinal mucosa from IBD patients. Gene expression of DRD2 and DRD3 from bulk RNA sequencing data in ileum and colon biopsies of healthy controls (HCs; *n* = 48), non-inflamed (non Inf) and inflamed (Inf) mucosa from patients with Crohn’s disease (CD; non Inf = 162, Inf = 73), and ulcerative colitis (UC; non Inf = 145, Inf = 121). The *Y*-axis on the graphs represents normalised transcript counts, obtained through TMM normalisation of raw counts from HTSeq. These normalised values are used to depict relative gene expression levels, enabling comparison across samples. Data are presented as medians and interquartile ranges. *, *p* < 0.05; ***, *p* < 0.001; ****, *p* < 0.0001 by Kruskal–Wallis’s test.

**Figure 4 ijms-26-10069-f004:**
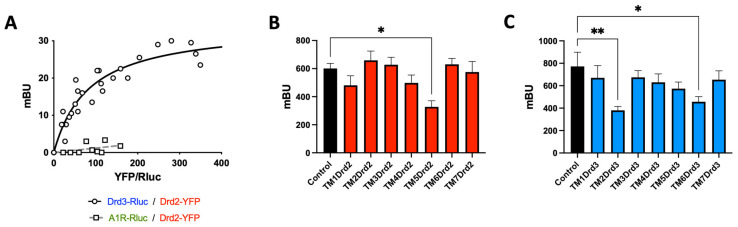
Drd2 and Drd3 interact physically through TM5 from Drd2 and TM2 and TM6 from Drd3. (**A**) HEK293T-cells were transfected with a constant amount of cDNA encoding for Renilla luciferase (RLuc) fused to Drd3 (as donor) or to adenosine receptor A1 (A1R, as a negative control) and increasing amounts of cDNA codifying for the Yellow Fluorescent Protein (YFP) fused to Drd2 (as the acceptor). BRET was expressed as milli BRET units (mBU) relative to the ratio between YFP fluorescence and RLuc activity. Data from five independent experiments is shown. (**B**,**C**) HEK293T-cells were transfected with Drd2-nYFP and Drd3-cYFP and a BiFC assay was performed. After 48 h, cells were left without treatment (control, black bar) or incubated with different TM peptides (0.4 μM; see Table 1) from Drd2 (**B**) or from Drd3 (**C**) for 4 h and YFP-associated fluorescence was determined. Mean ± SEM (*n* = 16–22 in B; *n* = 18–36 in (**C**)). *, *p* < 0.05; **, *p* < 0.01 by one-way ANOVA followed by Bonferroni’s post hoc test (**B**,**C**).

**Figure 5 ijms-26-10069-f005:**
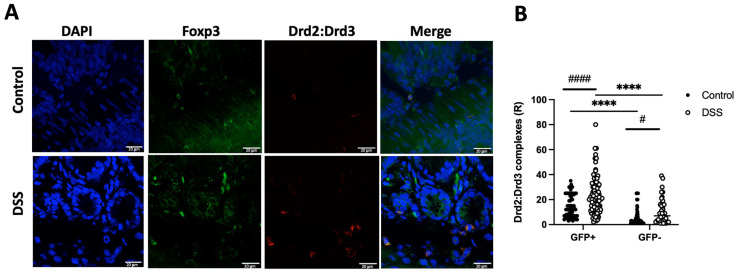
The Drd2:Drd3 heteromer is expressed on colonic Treg and it is up-regulated upon gut inflammation. *Foxp3^gfp^* mice were treated with 1.75% DSS or normal water (control) for 8 d and the extent of Drd2:Drd3 complexes on GFP^+^ cells (Treg) and GFP^−^ cells was determined in the colon by in situ PLA (*n* = 5 mice per group). (**A**) Representative images showing staining of nuclei (DAPI, blue), GFP (Foxp3, green), Drd2:Drd3 complexes (PLA, red) and merged images. Bar, 20 μm. (**B**) Quantification of the density of Drd2:Drd3 complexes on GFP^+^ and GFP^−^ cells with a lymphoid morphology. Values are the number of red dots per cell (R). Each symbol represents data obtained from an individual determination from 40 to 85 fields per group. ****, *p* < 0.0001 by two-way ANOVA followed by Sidak’s post hoc test. * indicates differences between GFP^+^ and GFP^−^ cells, while # (*p* < 0.05), or #### (*p* < 0.0001) indicates differences between treatments (control and DSS).

**Figure 6 ijms-26-10069-f006:**
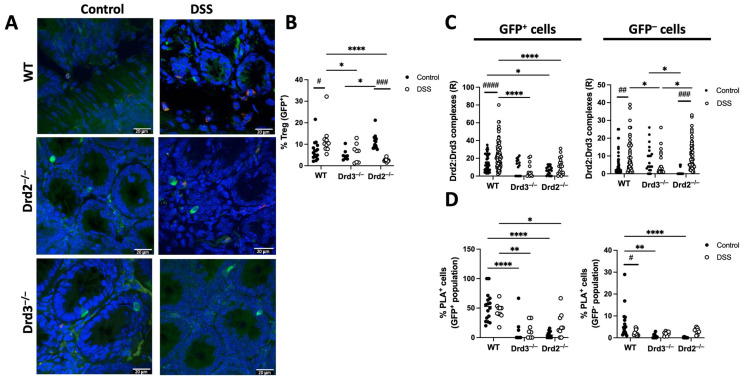
The population of colonic Treg increases upon gut inflammation, which is dependent on Drd2 and Drd3. *Drd2^f/f^*/*CD4^Cre^* (Drd2^−/−^), *Drd3*^−/−^ (Drd3^−/−^) or *Drd2*^+/+^*Drd3*^+/+^ (WT) *Foxp3^gfp^* mice were treated with 1.75% DSS or normal water (control) for 8 d and the extent of Treg infiltration and the expression of Drd2:Drd3 complexes on GFP^+^ cells (Treg) and GFP^−^ cells was determined in the colon by in situ PLA (*n* = 5 mice per group). (**A**) Representative images showing merged staining of nuclei (DAPI, blue), GFP (Foxp3, green), and Drd2:Drd3 complexes (PLA, red). Bar, 20 μm. (**B**) Quantification of the percentage of GFP+ cells from the total cells with lymphoid morphology. (**C**) Quantification of the density of Drd2:Drd3 complexes on GFP^+^ and GFP^−^ cells with lymphoid morphology. Values are the number of red dots per cell (R). (**D**) Quantification of the percentage of cells showing PLA+ staining on the GFP^+^ and GFP^−^ cell population with lymphoid morphology. Each symbol represents data obtained from an individual determination. *, *p* < 0.05; **, *p* < 0.01; ****, *p* < 0.0001 by two-way ANOVA followed by Sidak’s post hoc test. *, **, or **** indicate differences between genotypes, while # (*p* < 0.05), ## (*p* < 0.01), ### (*p* < 0.001), or ##### (*p* < 0.0001) indicate differences between treatments (control and DSS).

**Figure 7 ijms-26-10069-f007:**
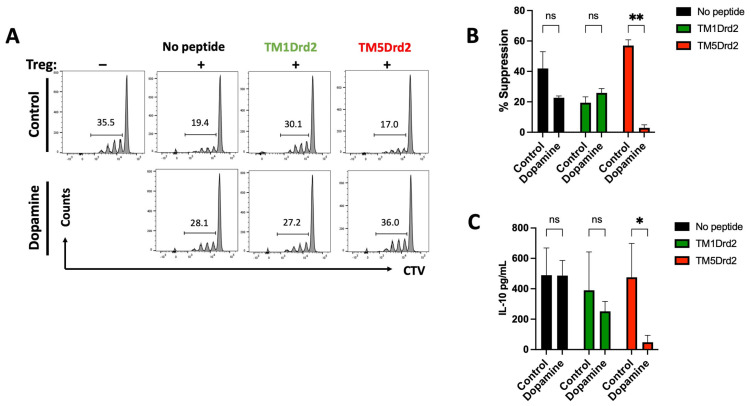
Disassembling the Drd2:Drd3 heteromer impairs the suppressive activity of Treg. Splenic Treg cells (CD4^+^GFP^+^) isolated from *CD45.2*^+/+^
*Foxp3^gfp^* mice were incubated with 4 μM TM1Drd2 (green) or TM5Drd2 (red) peptides for 4 h. During the last 30 min, cells were non-treated or treated with dopamine 2 μM. Naive CD4^+^ CD25^−^ T-cells (T naive) isolated from WT *CD45.1*^+/+^ mice were loaded with 5 μM CTV and activated with anti-CD3 and anti-CD28 Abs in the presence of peptide-treated Treg at a Tnaive/Treg ratio of 2:1. After 72 h, the extent of T naive proliferation was determined as the dilution of CTV-associated fluorescence in the CD4^+^ CD45.1^+^ GFP^−^ ZAq^−^ population by flow cytometry. (**A**) Representative dot plots. The marker shows proliferating T-cells. Numbers on the histogram indicate the percentage of proliferating cells. (**B**) Quantification was determined as the % of suppression (the percentage of reduction in the proliferation relative to the proliferation of T naive in the absence of Treg). (**C**) Quantification of IL-10 concentration in the culture supernatant by ELISA. (**B**,**C**) Values are the mean ± SEM from a representative experiment conducted in triplicate. Data from one out of three independent experiments are shown. *, *p* < 0.05; **, *p* < 0.01 by two-way ANOVA followed by Sidak’s post hoc test. ns, non-significant.

**Figure 8 ijms-26-10069-f008:**
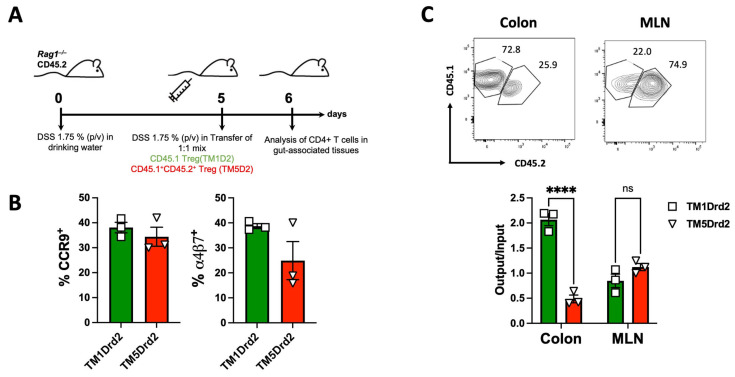
The disruption of Drd2:Drd3 heteromer assembly dampens the recruitment of Treg into the colonic mucosa upon inflammation. Splenic Treg cells (CD4^+^GFP^+^) isolated from *CD45.1*^+/+^
*Foxp3^gfp^* or from *CD45.1*^+/−^
*CD45.2*^+/−^
*Foxp3^gfp^* mice were incubated with 4 μM TM1Drd2 (green bars) or TM5Drd2 (red bars) peptides, respectively, for 4 h. Afterwards, both groups of Treg cells were mixed in 1:1 ratio (input) and then i.v. injected (7 × 10^5^ total cells per mouse) into *Rag1*^−/−^ recipient mice that previously received DSS for 5 days. Mice were further treated with 1.75% DSS for 24 h after T-cell transfer and then were sacrificed and the relative composition (CD45.1^+^ versus CD45.1^+^ CD45.2^+^) of GFP^+^ Treg isolated from the colonic lamina propria or MLNs was analysed. (**A**) Scheme illustrating the experimental strategy. (**B**) CCR9 and α4β7 expression was determined in the input and analysed by flow cytometry. (**C**) Top panels showing representative contour plots of donor Treg infiltrated into the colonic lamina propria and MLNs of recipients. Numbers indicate the percentage of cells in each region. Bottom panel shows the quantification of the relative abundance of CD45.1^+^ and CD45.1^+^ CD45.2^+^ Treg in particular tissues. Data is the % of CD45.1^+^ or CD45.1^+^ CD45.2^+^ donor cells in each tissue normalised by the percentage of those cells present in the input. (**B**,**C**) Each symbol represents data obtained from an individual mouse. Mean ± SEM are indicated. Data from a representative experiment are shown. ****, *p* < 0.0001 by two-way ANOVA followed by Sidak’s post hoc test.

**Table 1 ijms-26-10069-t001:** Peptides analogous to transmembrane segments of DRD2 and DRD3.

Functional Name	Short Name	Sequence ^1^
TM1hDRD2-TAT	TM1Drd2	ATLLTLLIAVIVFGNVLVSMAVS**YGRKKRRQRRR**
TAT-TM2hDRD2	TM2Drd2	**RRRQRRKKRGY**YLIVSLAVADLLVATLVMPWVVY
TM3hDRD2-TAT	TM3Drd2	IFVTLDVMMSTASILNLSAISI**YGRKKRRQRRR**
TAT-TM4hDRD2	TM4Drd2	**RRRQRRKKRGY**VTVMISIVWVLSFTISSPLLF
TM5hDRD2-TAT	TM5Drd2	FVVYSSIVSFYVPFIVTLLVYIKIY**YGRKKRRQRRR**
TAT-TM6hDRD2	TM6Drd2	**RRRQRRKKRGY**MLAIVLGVFIISWLPFFITHIL
TM7hDRD2-TAT	TM7Drd2	AFTWLGYVNSAVNPIIYTTFNI**YGRKKRRQRRR**
TM1hDRD3-TAT	TM1Drd3	ALSYSALILAIVFGNGLVSMAVL**YGRKKRRQRRR**
TAT-TM2hDRD3	TM2Drd3	**RRRQRRKKRGY**YLVVSLAVADLLVATLVMPWVVY
TM3hDRD3-TAT	TM3Drd3	VFVTLDVMMSTASILNLSAISI**YGRKKRRQRRR**
TAT-TM4hDRD3	TM4Drd3	**RRRQRRKKRGY**VALMITAVWVLAFAVSSPLLF
TM5hDRD3-TAT	TM5Drd3	FVIYSSVVSFYLPFGVTVLVYARIY**YGRKKRRQRRR**
TAT-TM6hDRD3	TM6Drd3	**RRRQRRKKRGY**MVAIVLGAFIVSWLPFFLTHVL
TM7hDRD3-TAT	TM7Drd3	ATTWLGYVNSALNPVIYTTFNI**YGRKKRRQRRR**

^1^ Predicted transmembrane regions were obtained for human DRD2 (code P14416) or human DRD3 (code P35462) from uniport. To ensure the proper delivery of transmembrane peptides with the correct orientation in the plasma membrane, the TAT peptide (indicated in bold) was added in a direct orientation (YGRKKRRQRRR) in the C-terminal of odd transmembrane segments and in the inverse orientation (RRRQRRKKRGY) in the N-terminal of even transmembrane segments. The TAT peptide is a cell-penetrating peptide derived from the transactivator of transcription protein of the human immunodeficiency virus. In addition, to avoid the formation of bisulfide bridges, cysteines were replaced with serines (underlined).

**Table 2 ijms-26-10069-t002:** Histological scoring calculation.

Feature	Score	Description
Inflammation	0	None
	1	Slight
	2	Moderate
	3	Severe
Extent of injury	0	None
	1	Mucosal
	2	Mucosal and sub-mucosal
	3	Transmural
Crypt damage	0	None
	1	Basal 1/2 damage
	2	Basal 2/3 damage
	3	Only surface epithelium intact
	4	Entire crypt and epithelium lost
The score of each parameter above is multiplied by a factor reflecting the percentage of tissue involved as indicated below		
Percentage involved	Factor	
0–25%	1	
26–50%	2	
51–75%	3	
76–100%	4	

## Data Availability

Raw data from bulk RNA transcriptomic analysis shown in Figure 3 are available at [33]. The data underlying other figures will be shared on reasonable request to the corresponding author.

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
