# Peer review of "The Heteromeric Dopamine Receptor D2:D3 Controls the Gut Recruitment and Suppressive Activity of Regulatory T-Cells"

_ijms, 2025, doi:10.3390/ijms262010069_

Round 1

Reviewer 1 Report

Comments and Suggestions for Authors

Inflammatory bowel disease (IBD) exerts a significant impact on the quality of life of millions of individuals worldwide whose etiology has not been elucidated. Little is also known about the role of dopamine and dopamine receptor in the pathogenesis of IBD. This manuscript mainly investigated the role of heteromeric dopamine receptor D2:D3 in the gut recruitment and suppressive activity of regulatory T-cells, and their results provided an important reference value for basic research in gut inflammation as well as an attractive molecular target for translational research in IBD. The references in this MS are appropriate.

Although the topic is interesting, and the subject in this MS is worthy of investigation, below some suggestions may be helpful for shaping this manuscript.

Major revisions:

  1. In Figure 1, it’s suggested that light microscopy of the effect of Drd2-agonist on histologic changes in the colon during DSS-induced colitis in mice (H&E staining) should be provided, as macroscopic and histologic results can be served as an important supplement in this MS.

This MS reported that the DRD2 and DRD3 transcripts levels obtained from intestinal mucosa of IBD patients and healthy controls were compared in Figure 3, and in Figure 5, the Drd2:Drd3 heteromer was expressed on colonic Treg and up-regulated upon gut inflammation in mice. It’s suggested that the results of the DRD2 and DRD3 protein and Drd2:Drd3 heteromer expression in IBD patients could be more valuable. Additionally, in Figure 5 and Figure 6, clearer size mark (20 μm?) on the scale bar should be provided.

  1. This MS indicated that the D2:D3 heteromer disassembly dampened the suppressive Treg activity and impaired their colonic tropism upon inflammation. It’s suggested that the authors provide the results of downstream signaling pathways involved in these processes, or add some relevant content in the discussion section.

  1. It’s suggested that proposed scheme on the role of heteromeric dopamine receptor D2:D3 in the gut recruitment and suppressive activity of regulatory T-cells under inflammatory conditions should be provided.

Additionally, conclusions section should be added according to the journal’s requirements.

  1. In Materials and Methods section, it’s suggested that reagents section should be added, such as sumanirole and antibodies for flow cytometry were purchased from? The peptide analogues to transmembrane (TM) segments were synthesized by? Additionally, what were used as controls in flow cytometry analysis? Was this study approved by the institutional research ethics committee, and written informed consent obtained from each subject as human samples were used? If yes, please provide the relevant materials.

Minor revisions:

  1. Please refer to recent papers published in International Journal of Molecular Sciences and correct the format, such as:

In page 1, line 8, “Huechuraba (8580702)” should be changed to “Huechuraba 8580702”, and all author’s name should be provided after his/her email address.

In page 4, line 148, “Fig. 1A” should be changed to “Figure 1A”. Other places should be modified accordingly.

In line 193, p in p<0.05 should be in italic. Other places should be modified accordingly.

In line 197, “…Treg activity we wondered...” should be changed to “…Treg activity, we wondered...”.

In line 222, line 434, 5 in the“7x105” should be displayed in superscript.

In line 229, line 444, “Mean  SEM” should be changed to “Mean ± SEM”.

In line 322-325, “Fig. 5A-B” should be changed to “Figure 5”.

In line 315, 328, 362, “1,75% DSS” should be changed to “1.75% DSS”.

In line 438, α4ß7 should not be bold.

In line 424, 426, “these results indicate that” is repeated.

  1.  In all references, if there are more than ten authors, list the first ten authors and then use et al; all issue should be deleted; all page range should be provided.

According to the journal’s requirements, all references format should be:

  1. Author 1, A.B.; Author 2, C.D. Title of the article. Abbreviated Journal Name Year, Volume, page range.

Please check all references including content and format carefully.

Author Response

REVIEWER 1

Comments and Suggestions for Authors

Inflammatory bowel disease (IBD) exerts a significant impact on the quality of life of millions of individuals worldwide whose etiology has not been elucidated. Little is also known about the role of dopamine and dopamine receptor in the pathogenesis of IBD. This manuscript mainly investigated the role of heteromeric dopamine receptor D2:D3 in the gut recruitment and suppressive activity of regulatory T-cells, and their results provided an important reference value for basic research in gut inflammation as well as an attractive molecular target for translational research in IBD. The references in this MS are appropriate.

Although the topic is interesting, and the subject in this MS is worthy of investigation, below some suggestions may be helpful for shaping this manuscript.

Major revisions:

  1. In Figure 1, it’s suggested that light microscopy of the effect of Drd2-agonist on histologic changes in the colon during DSS-induced colitis in mice (H&E staining) should be provided, as macroscopic and histologic results can be served as an important supplement in this MS.

ANSWER: We have included the histological analysis of mice treated or not with the Drd2-agonist upon DSS-induced colitis. Despite this analysis did not show differences between colitis mice treated or not with the drug, the analysis of body weight loss still suggests that the systemic administration of the Drd2-agonist exert a beneficial effect attenuating the disease manifestation. These results have been included in the new version of the paper (Figure 1D), and commented in the section of results (lines 323-324).

This MS reported that the DRD2 and DRD3 transcripts levels obtained from intestinal mucosa of IBD patients and healthy controls were compared in Figure 3, and in Figure 5, the Drd2:Drd3 heteromer was expressed on colonic Treg and up-regulated upon gut inflammation in mice. It’s suggested that the results of the DRD2 and DRD3 protein and Drd2:Drd3 heteromer expression in IBD patients could be more valuable.

ANSWER: Please note that results shown in figure 3 correspond to analyses of bulk RNAseq from biopsies obtained from intestinal mucosa of IBD patients and healthy controls from a previous study (reference 33). We do not have access to those biopsies, which were fully used for RNAseq purposes.

Additionally, in Figure 5 and Figure 6, clearer size mark (20 μm?) on the scale bar should be provided.

ANSWER: Clearer size mark has been added to the representative microscopy images in figures 5 and 6.

  1. This MS indicated that the D2:D3 heteromer disassembly dampened the suppressive Treg activity and impaired their colonic tropism upon inflammation. It’s suggested that the authors provide the results of downstream signaling pathways involved in these processes, or add some relevant content in the discussion section.

ANSWER: We thank the reviewer for this important observation. We have added new results showing the crosstalk of the Drd2:Drd3 heteromer at the level of cAMP production, ERK1/2 phosphorylation and AKT phosphorylation. These new results are presented in the new figures S1 and S2, and comments in the section of Results and Discussion.

Results section (lines 584-590): To analyse the crosstalk of Drd2 and Drd3 at the level of signalling pathways activated, we determined the ability of stimulating these receptors to modulate cAMP production. Both Drd2 and Drd3 stimulation were coupled with the inhibition of cAMP production, and the simultaneous stimulation synergises on it (Figure S1). In addition, we analysed the potential coupling of Drd2 and Drd3 to the activation of MAPK and AKT pathways. The results show that both Drd2 and Drd3 induced the phosphorylation of ERK1/2 and of AKT, however their effects were not additive (Figure S2).

Discussion section (lines 1046-1057): Since the stimulation of Drd2 and Drd3 triggers opposite functions in Treg cells, including gut tropism and suppressive activity, we aimed to identify signalling pathways coupled to Drd2 and Drd3 in opposite directions. However, we found that cAMP, ERK1/2 phosphorylation, and AKT phosphorylation were all coupled to Drd2 and Drd3 stimulation in the same direction (Figures S1 and S2). Since CD25 expression was significantly affected by the Drd2:Drd3 heteromer disruption (Figure S3), and blimp-1 negatively regulates CD25 expression on Treg (50), we also evaluated how blim-1 expression is affected by the heteromer integrity; however we did not find significant effects (Figure S3). Future research should aim to understand the molecular mechanisms by which Drd2 and Drd3 stimulation in the heteromer trigger opposite functional effects at the level of migration to the gut and suppressive activity.

  1. It’s suggested that proposed scheme on the role of heteromeric dopamine receptor D2:D3 in the gut recruitment and suppressive activity of regulatory T-cells under inflammatory conditions should be provided.

 ANSWER: We have included a new figure (Figure S4), which illustrates the proposed model for the function of Drd2:Drd3 heteromer in Treg in the context of gut inflammation.

Additionally, conclusions section should be added according to the journal’s requirements.

 ANSWER: We have added a section of conclusions (section 4) as recommended by the reviewer.

  1. In Materials and Methods section, it’s suggested that reagents section should be added, such as sumanirole and antibodies for flow cytometry were purchased from? The peptide analogues to transmembrane (TM) segments were synthesized by? Additionally, what were used as controls in flow cytometry analysis? Was this study approved by the institutional research ethics committee, and written informed consent obtained from each subject as human samples were used? If yes, please provide the relevant materials.

 ANSWER: We have added a section (section 5.2.) of reagents in Materials and Methods, in the new versión of the paper.

At the end of the section 5.8 “Flow Cytometry Analysis” we have stated that “All analyses were controlled in a first step with the autofluorescence and in a second step with Fluorescence Minus One (FMO)”.

Institutional ethical approval statement for animal experimentaion is indicated in lines 1525-1528:

Institutional Review Board Statement: All procedures and housing of mice were compliant with the recommendations in the 8th edition of the Guide for the Care and Use of Laboratory Animals and with the United States Public Health Service Policy. The protocol was approved by the IACUC of Fundación Ciencia & Vida (Permit Number: P016-2020).

Regarding to the experimentation with human samples: Please note that results shown in figure 3 correspond to analyses of bulk RNAseq from biopsies obtained from intestinal mucosa of IBD patients and healthy controls from a previous study (reference 33). So, we did not require ethical approval for this in the present study.

Minor revisions:

  1. Please refer to recent papers published in International Journal of Molecular Sciences and correct the format, such as:

In page 1, line 8, “Huechuraba (8580702)” should be changed to “Huechuraba 8580702”, and all author’s name should be provided after his/her email address.

ANSWER: Postal codes from all affiliation addresses have been left without parenthesis.Authors names have been specificied after email addresses.

In page 4, line 148, “Fig. 1A” should be changed to “Figure 1A”. Other places should be modified accordingly.

ANSWER: We have spelled “Figure x” instead “Fig. x” in all places throughout the whole manuscript.

In line 193, p in p<0.05 should be in italic. Other places should be modified accordingly.

ANSWER: “p” indicating probabilities have been changed to “p” in italic in the whole manuscript.

In line 197, “…Treg activity we wondered...” should be changed to “…Treg activity, we wondered...”.

ANSWER: we have included this change in the new version of the manuscript.

In line 222, line 434, 5 in the“7x105” should be displayed in superscript.

ANSWER: The number 5 has been written as superscript now.

In line 229, line 444, “Mean  SEM” should be changed to “Mean ± SEM”.

ANSWER: We have corrected these typos.

In line 322-325, “Fig. 5A-B” should be changed to “Figure 5”.

ANSWER: We have changed “Fig. 5A-B” with “Figure 5”.

In line 315, 328, 362, “1,75% DSS” should be changed to “1.75% DSS”.

ANSWER: “1,75% DSS” has been replaced with “1.75% DSS” throughout the whole manuscript.

In line 438, α4ß7 should not be bold.

ANSWER: it has been corrected.

In line 424, 426, “these results indicate that” is repeated.

 ANSWER: This has been corrected in the new version of the paper.

  1.  In all references, if there are more than ten authors, list the first ten authors and then use et al; all issue should be deleted; all page range should be provided.

According to the journal’s requirements, all references format should be:

  1. Author 1, A.B.; Author 2, C.D. Title of the article. Abbreviated Journal Name YearVolume, page range.

Please check all references including content and format carefully.

ANSWER: We have updated the citations style with the MDPI EndNote style indicated in the Instructions for Authors in the web page of International Journal of Molecular Sciences.

Reviewer 2 Report

Comments and Suggestions for Authors

~ Please rewrite the abstract with tangible results 

~ Please rewrite the inroduction to justify the novelty of this work. I can not get the novelty of this work. In my opinion there is no novelty.

~Mean SEM indicated I think should be Mean ± SEM

~Table 1. The sequence for TAT-TM2hDRD2 and  TAT-TM2hDRD3 are identical, please verify

~ Please revise the whole manuscript and corect typhographical errors. 

~The quality of all figures are extreamely weak, please try to improve it

~Critically revise the manuscript for scientific errors and Greek letters error 

~Critically revise the manuscript for typographiscal mistakes 

Author Response

REVIEWER 2

Comments and Suggestions for Authors

~ Please rewrite the abstract with tangible results 

ANSWER: The abstract has been rewritten with tangible results.

~ Please rewrite the inroduction to justify the novelty of this work. I can not get the novelty of this work. In my opinion there is no novelty.

ANSWER: We have highlighted the novelty of our findings at the end of the introduction.

~Mean SEM indicated I think should be Mean ± SEM

~Table 1. The sequence for TAT-TM2hDRD2 and  TAT-TM2hDRD3 are identical, please verify.

ANSWER: We have verified the sequences in table 1, and they are right. Please note that they are not identical: sequence from TM2Drd2 (excluding the TAT peptide) is YLIVSLAVADLLVATLVMPWVVY, whereas TM2Drd3 sequence is YLVVSLAVADLLVATLVMPWVVY. The third amino acid (underlined) is different. Despite they are not identical, they are, indeed, very similar, according to the high degree of sequence homology between Drd2 and Drd3.

~ Please revise the whole manuscript and corect typhographical errors. 

ANSWER: Typos have been revised throughout the whole manuscript.

~The quality of all figures are extreamely weak, please try to improve it

ANSWER: Figures have been generated with higher resolution in the new version of the paper.

~Critically revise the manuscript for scientific errors and Greek letters error 

ANSWER: Greek symbols have been revised and corrected throughout the whole manuscript.

~Critically revise the manuscript for typographiscal mistakes 

ANSWER: Typos have been revised throughout the whole manuscript.

Reviewer 3 Report

Comments and Suggestions for Authors

Review-MDPI-IJMS-3840451

Title: The heteromeric dopamine receptor D2:D3 controls the gut recruitment and suppressive activity of regulatory T-cells

Comments:

In current manuscript Mora et al. investigated the role of dopaminergic signaling in the regulation of Treg function during intestinal inflammation. The authors provide a compelling set of genetic, pharmacologic, and in vitro functional data to demonstrate that Drd2 activation enhances Treg suppressive activity and colonic homing, in contrast to Drd3 stimulation, which was previously shown to have the opposite effect. They identify a Drd2:Drd3 heteromer as a novel dopamine-sensing complex that regulates Treg functionality in the gut. The study is timely and has a significant conceptual advancement in the field of neuroimmune interactions and mucosal immunology. The following suggestions could improve the manuscript readability and understandings for broader audience.

  1. Some terms, for instance, "colonic tropism, gut tropism", may not be familiar to a broader audience and would be better to explain briefly.
  2. The identification of a Drd2:Drd3 heteromer is novel, but the mechanistic consequences of its disruption need further explanation. For instance;
  3. How does heteromer disassembly alter downstream signaling in Tregs?
  4. Do Drd2 and Drd3 have opposing intracellular effects when acting independently?

This could be improved by even speculative discussion or additional signaling analysis (e.g., STAT5, Foxp3 expression) which would enhance mechanistic insight.

  1. Throughout the manuscript the authors focused on in vivo data based on CD4⁺ T cell-specific Drd2 deletion, but conclusions focus on Tregs.
  • Consider including Treg-specific deletion models (e.g., using Foxp3-Cre) or clarify in discussion that some effects may be due to broader CD4⁺ T cell populations.
  • Alternatively, flow cytometric gating on Foxp3⁺ subsets in functional assays would strengthen the conclusion.
  1. The results are informative but quite dense. Breaking into smaller thematic paragraphs would improve readability.
  2. A schematic model summarizing the proposed role of the Drd2:Drd3 heteromer in Treg modulation would be helpful for readers.

Author Response

REVIEWER 3

Comments and Suggestions for Authors

Review-MDPI-IJMS-3840451

Title: The heteromeric dopamine receptor D2:D3 controls the gut recruitment and suppressive activity of regulatory T-cells

Comments:

In current manuscript Mora et al. investigated the role of dopaminergic signaling in the regulation of Treg function during intestinal inflammation. The authors provide a compelling set of genetic, pharmacologic, and in vitro functional data to demonstrate that Drd2 activation enhances Treg suppressive activity and colonic homing, in contrast to Drd3 stimulation, which was previously shown to have the opposite effect. They identify a Drd2:Drd3 heteromer as a novel dopamine-sensing complex that regulates Treg functionality in the gut. The study is timely and has a significant conceptual advancement in the field of neuroimmune interactions and mucosal immunology. The following suggestions could improve the manuscript readability and understandings for broader audience.

  1. Some terms, for instance, "colonic tropism, gut tropism", may not be familiar to a broader audience and would be better to explain briefly.

ANSWER: The terms “colonic tropism” and “gut tropism” have been avoided in the abstract and explained in the introduction (line 232).

  1. The identification of a Drd2:Drd3 heteromer is novel, but the mechanistic consequences of its disruption need further explanation. For instance;

How does heteromer disassembly alter downstream signaling in Tregs?

Do Drd2 and Drd3 have opposing intracellular effects when acting independently?

This could be improved by even speculative discussion or additional signaling analysis (e.g., STAT5, Foxp3 expression) which would enhance mechanistic insight.

ANSWER: We thank the reviewer for this important observation. We have added new results showing the crosstalk of the Drd2:Drd3 heteromer at the level of cAMP production, ERK1/2 phosphorylation and AKT phosphorylation. These new results are presented in the new figures S1 and S2, and comments in the section of Results and Discussion.

Results section (lines 584-590): To analyse the crosstalk of Drd2 and Drd3 at the level of signalling pathways activated, we determined the ability of stimulating these receptors to modulate cAMP production. Both Drd2 and Drd3 stimulation were coupled with the inhibition of cAMP production, and the simultaneous stimulation synergises on it (Figure S1). In addition, we analysed the potential coupling of Drd2 and Drd3 to the activation of MAPK and AKT pathways. The results show that both Drd2 and Drd3 induced the phosphorylation of ERK1/2 and of AKT, however their effects were not additive (Figure S2).

Discussion section (lines 1046-1057): Since the stimulation of Drd2 and Drd3 triggers opposite functions in Treg cells, including gut tropism and suppressive activity, we aimed to identify signalling pathways coupled to Drd2 and Drd3 in opposite directions. However, we found that cAMP, ERK1/2 phosphorylation, and AKT phosphorylation were all coupled to Drd2 and Drd3 stimulation in the same direction (Figures S1 and S2). Since CD25 expression was significantly affected by the Drd2:Drd3 heteromer disruption (Figure S3), and blimp-1 negatively regulates CD25 expression on Treg (50), we also evaluated how blim-1 expression is affected by the heteromer integrity; however we did not find significant effects (Figure S3). Future research should aim to understand the molecular mechanisms by which Drd2 and Drd3 stimulation in the heteromer trigger opposite functional effects at the level of migration to the gut and suppressive activity.

  1. Throughout the manuscript the authors focused on in vivo data based on CD4⁺ T cell-specific Drd2 deletion, but conclusions focus on Tregs.
  • Consider including Treg-specific deletion models (e.g., using Foxp3-Cre) or clarify in discussion that some effects may be due to broader CD4⁺ T cell populations.
  • Alternatively, flow cytometric gating on Foxp3⁺ subsets in functional assays would strengthen the conclusion.
  • ANSWER: We have modified the conclusions commented about the results obtained with Drd2flox/CD4Cre mice as indicated in Lines 323-325: All these results suggest that Drd2-stimulation on CD4+ T-cells dampens their inflammatory function and favours the production of the anti-inflammatory cytokine IL-10 (Figure 1 A-C).
  • Please note that many other results actually allow to conclude about the effect of Drd2 or the Drd2:Drd3 heteromer on Treg cells:
  • Figure 1E: Are Treg in vitro treated with Drd2-agonist and then exposed to suppressive assays.
  • Figure 2A: Are ex vivo Treg (wild type or Drd2 -/-) transferred into WT recipients treated with DSS; Figure 2B and C are Treg (wild type or Drd2 -/-) in vitro.
  • Figure 7: Are Treg in vitro treated with dopamine and peptides and then evaluated in suppressive assays.
  • Figure 8: Are ex vivo Treg (treated with peptides ) transferred into WT recipients treated with DSS.
  •  
  1. The results are informative but quite dense. Breaking into smaller thematic paragraphs would improve readability.

ANSWER: We have broken the section of results in smaller thematic paragraphps to improve readability, as recommended by the reviewer.

  1. A schematic model summarizing the proposed role of the Drd2:Drd3 heteromer in Treg modulation would be helpful for readers.

ANSWER: We have included a new figure (Figure S4), which illustrates the proposed model for the function of Drd2:Drd3 heteromer in Treg in the context of gut inflammation.

Round 2

Reviewer 1 Report

Comments and Suggestions for Authors

My previous comments have been reasonably responded. However, below some revisions are still required, such as:

1. In “5.4. Histological analysis” section, it is recommended to provide the calculation formula of histopathological score of colonic tissue in the revised MS in detail. 

  1. In page 1, line 7-31, all author’s name should be abbreviated and provided after his/her email address with ().

In page 1, line 33-34, “Correspondence: Rodrigo Pacheco, Avenida Del Valle Norte #725, Huechuraba 8580702 Santiago, Chile. 33 ORCID: 0000-0001-8057-9806. E-mail: rpacheco@cienciavida.org; rodrigo.pacheco@uss.cl.” should be changed to “Correspondence: rpacheco@cienciavida.org(R.P.)”.

Please refer to recent papers published in International Journal of Molecular Sciences and correct the format.

3. In reference 1, “PONE-D-12-29699 [pii].” should be deleted.In reference 18, “nature11748 [pii]. ” should be deleted.

In reference 5, 12, 17, 33, 39, 54, 58, 59, all journal name should be abbreviated, and add ”.” after them.

Please check other references including content and format carefully according to the journal’s requirements.

 4. Where is Figure S1-4?

Author Response

  1. In “5.4. Histological analysis” section, it is recommended to provide the calculation formula of histopathological score of colonic tissue in the revised MS in detail. 

ANSWER: We thank the reviewer for this observation. We have included a new table (Table 2) in the section 5.4, in which is detailed how to calculate the histological score.

  1. In page 1, line 7-31, all author’s name should be abbreviated and provided after his/her email address with ().

In page 1, line 33-34, “Correspondence: Rodrigo Pacheco, Avenida Del Valle Norte #725, Huechuraba 8580702 Santiago, Chile. 33 ORCID: 0000-0001-8057-9806. E-mail: rpacheco@cienciavida.org; rodrigo.pacheco@uss.cl.” should be changed to “Correspondence: rpacheco@cienciavida.org(R.P.)”.

Please refer to recent papers published in International Journal of Molecular Sciences and correct the format.

 ANSWER: We have corrected the format indicated by the reviewer.

  1. In reference 1, “PONE-D-12-29699 [pii].” should be deleted.In reference 18, “nature11748 [pii]. ” should be deleted.

In reference 5, 12, 17, 33, 39, 54, 58, 59, all journal name should be abbreviated, and add ”.” after them.

Please check other references including content and format carefully according to the journal’s requirements.

 ANSWER: We have corrected the format indicated by the reviewer.

  1. Where is Figure S1-4?

ANSWER: We uploaded a PDF file with the supplementary figures S1-S4 as a "figure file" in the MDPI platform. To be sure you receive the figures, we are also submitting this PDF file to Mr. Caden Wang's email (Assistant Editor).
